# PROXIMAL DIFFUSION NEURAL SAMPLER

**Wei Guo**[*], **Jaemoo Choi**[*], **Yuchen Zhu, Molei Tao**[†], **Yongxin Chen**[‡]
Georgia Institute of Technology
{wei.guo, jchoi843, yzhu738, mtao, yongchen}@gatech.edu

## ABSTRACT

The task of learning a diffusion-based neural sampler for drawing samples from an unnormalized target distribution can be viewed as a stochastic optimal control problem on path measures. However, the training of neural samplers can be challenging when the target distribution is multimodal with significant barriers separating the modes, potentially leading to mode collapse. We propose a framework named **Proximal Diffusion Neural Sampler (PDNS)** that addresses these challenges by tackling the stochastic optimal control problem via proximal point method on the space of path measures. PDNS decomposes the learning process into a series of simpler subproblems that create a path gradually approaching the desired distribution. This staged procedure traces a progressively refined path to the desired distribution and promotes thorough exploration across modes. For a practical and efficient realization, we instantiate each proximal step with a proximal weighted denoising cross-entropy (WDCE) objective. We demonstrate the effectiveness and robustness of PDNS through extensive experiments on both continuous and discrete sampling tasks, including challenging scenarios in molecular dynamics and statistical physics. Our code is available at https://github.com/AlexandreGUO2001/PDNS.

## 1 INTRODUCTION

Sampling from an unnormalized Boltzmann distribution is a fundamental task across various domains, including computational statistics (Liu, 2008; Brooks et al., 2011), Bayesian inference (Gelman et al., 2013), statistical mechanics (Landau & Binder, 2014), etc. Formally, we aim to draw samples from a target distribution $\pi$ on the state space $\mathcal{X}$, whose probability density or mass function is specified by an energy function $E : \mathcal{X} \to \mathbb{R}$ and an inverse temperature $\beta > 0$:

$$\pi(x) = \frac{1}{Z} e^{-\beta E(x)}, \qquad \text{where} \quad Z := \int_{\mathcal{X}} e^{-\beta E(x)} dx \quad \text{or} \quad \sum_{x \in \mathcal{X}} e^{-\beta E(x)}. \tag{1}$$

Classical Markov chain Monte Carlo (MCMC) methods may suffer from slow mixing when in high dimensions or when the target distribution is multimodal (Łatuszyński et al., 2025; He & Zhang, 2025), motivating neural samplers that transport a known reference distribution to the target via score-based diffusion or normalizing flows (Gabrié et al., 2022; Zhang & Chen, 2022).

Recently, diffusion-based neural samplers have been extensively studied in learning high-dimensional complex distributions (Zhang & Chen, 2022; Vargas et al., 2023; Phillips et al., 2024; Havens et al., 2025; Liu et al., 2025). These approaches typically parameterize the control of a diffusion process and aim to transport a simple initial distribution toward a prescribed target distribution $\pi$ by solving a stochastic optimal control (SOC) problem. However, when the model initially captures only a few modes in the early stages of training, the global loss tends to reinforce those modes, thereby concentrating probability mass to only several modes, resulting in mode collapse. This observation motivates replacing one-shot global minimization with a sequence of gradual and constrained steps that improve control while preserving mode coverage.

In this paper, we propose the **Proximal Diffusion Neural Sampler (PDNS)**, a unified framework of diffusion-based sampling for both continuous and discrete domains, which leverages proximal

---

[*]Equal contribution. [† ‡] Joint mentorship. [‡] Corresponding author.

iterations over path measures. Starting from a given initial diffusion process, PDNS iteratively addresses a local sub-problem and updates the control of the dynamics so that the terminal marginal distribution progressively approaches the target distribution $\pi$. This proximal structure yields stable optimization and mitigates mode collapse in multi-modal settings. As a one instantiation of PDNS, we develop *proximal* algorithmic variants of cross-entropy (CE) and weighted denoising cross-entropy (WDCE) losses (Phillips et al., 2024; Zhu et al., 2025), which are widely adopted and computationally efficient. Finally, we evaluate our PDNS in the multi-modal and high-dimensional benchmarks for both continuous and discrete target distributions. We compare against recent works, including the original WDCE variant, achieving state-of-the-art for many benchmarks. Our experimental results show the effectiveness and robustness of our methods.

**Contributions** Our key contributions are summarized as follows. **1.** In Sec. 2, by using a path measure formulation, we integrate continuous and discrete SOC-based samplers into the unified framework. **2.** In Sec. 3, starting from this unified viewpoint, we demonstrate the problem of mode collapse in the training of diffusion neural samplers. Then, we introduce *Proximal Diffusion Neural Sampler (PDNS)*, which performs proximal steps in path space to stabilize learning and mitigate collapse. **3.** In Sec. 4, we develop **proximal weighted denoising cross-entropy (proximal WDCE)** for both continuous and discrete settings as a concrete example of PDNS. We then outline principled strategies for selecting the proximal step size. **4.** Finally, in Sec. 5, we provide extensive experiments across continuous and discrete sampling tasks, demonstrating that PDNS improves stability, mode coverage, and overall sampling quality relative to strong baselines.

**Notations** Let $\mathcal{X}$ be the state space and $T > 0$ be the terminal time. A path measure is a probability measure $\mathbb{P} : [0, T] \times \mathcal{X} \to \mathbb{R}$ on the space of all paths, which is a subset of all functions from $[0, T]$ to $\mathcal{X}$.[1] All path measures considered in the paper start from the same initial law $\mu$, i.e., $\mathbb{P}_0 = \mu$. We denote $\mathbb{P}^{\mathrm{ref}}$ as a reference path measure and denote its terminal marginal by $\nu := \mathbb{P}_T^{\mathrm{ref}}$. The target distribution is $\pi$ on the state space $\mathcal{X}$, whose probability density function (p.d.f.) or probability mass function (p.m.f.) is specified up to a normalizing constant $Z$ by an energy function $E : \mathcal{X} \to \mathbb{R}$ and inverse temperature $\beta > 0$, i.e., $\pi(x) \propto e^{-\beta E(x)}$. Expectation under a path measure and its marginals are written as $\mathbb{E}_{\mathbb{P}}[\cdot]$ and $\mathbb{E}_{\mathbb{P}_t}[\cdot]$, respectively. $1_A \in \{0, 1\}$ is the indicator of a statement or event $A$, which is 1 if.f. $A$ holds or is true. For $x = (x^1, ..., x^d) \in \mathbb{R}^d$, the notation $x^{i \leftarrow c}$ means the vector obtained by replacing the $i$-th entry of $x$ with $c$.

## 2 UNIFIED FRAMEWORK FOR SOC-BASED NEURAL SAMPLER

In this section, we discuss the key concepts and methods that form the foundation of our approach, including stochastic optimal control (SOC) interpretation of diffusion samplers (Zhang & Chen, 2022; Vargas et al., 2023; Domingo-Enrich et al., 2025; Liu et al., 2025; Choi et al., 2025) and the weighted cross-entropy-based approaches (Phillips et al., 2024; Zhu et al., 2025). We defer detailed discussions of the related works to Sec. A.

### 2.1 LEARNING NEURAL SAMPLER AS A STOCHASTIC OPTIMAL CONTROL PROBLEM

**Unifying Framework for SOC-based Neural Samplers** One approach to building diffusion samplers is to formulate them through a stochastic optimal control (SOC) perspective. We aim to fit a neural-network-parameterized path measure $\mathbb{P}^\theta$ to an optimal one $\mathbb{P}^*$ whose terminal distribution is the target $\pi$. Let $\mathbb{P}^{\mathrm{ref}} : [0, T] \times \mathcal{X} \to \mathbb{R}$ be a reference path measure[2] where $\mathbb{P}_0^{\mathrm{ref}} = \mu$ and $\mathbb{P}_1^{\mathrm{ref}} = \nu$. We assume that the reference path measure is **memoryless** (Domingo-Enrich et al., 2025): the initial and terminal distributions are independent, i.e.,

$$\mathbb{P}_{0,T}^{\mathrm{ref}}(X_0, X_T) = \mathbb{P}_0^{\mathrm{ref}}(X_0)\mathbb{P}_T^{\mathrm{ref}}(X_T) = \mu(X_0)\nu(X_T). \tag{2}$$

With this memoryless condition, the following holds:

---

[1] For instance, paths induced by Brownian motions are a.s. $C^{\frac{1}{2}-}$-Hölder continuous, and paths induced by continuous-time Markov chains are a.s. piecewise constant, right-continuous with left limits.

[2] Throughout this paper, path measure refers to the distribution of a stochastic process indexed by $t \in [0, T]$ and takes values on $\mathcal{X}$.

$$\mathbb{P}^* := \operatorname*{argmin}_{\mathbb{P}^\theta: \, \mathbb{P}_0^\theta = \mu} \left[ -\mathbb{E}_{\mathbb{P}^\theta(X)} \, r(X_T) + \mathrm{KL}(\mathbb{P}^\theta \| \mathbb{P}^{\mathrm{ref}}) \right] \tag{3}$$

$$\implies \mathbb{P}^*(X) = \frac{1}{Z} \mathbb{P}^{\mathrm{ref}}(X) \mathrm{e}^{r(X_T)}, \text{ where } r := -\beta E - \log \nu; \; \mathbb{P}_0^* = \mu, \; \mathbb{P}_T^* = \pi.$$

The proof is provided in Sec. C. In other words, the optimal path measure $\mathbb{P}^*$ bridges between the initial distribution of the reference path measure $\mathbb{P}^{\mathrm{ref}}$ to our target distribution $\pi$. We remark that the memoryless condition on $\mathbb{P}^{\mathrm{ref}}$ is necessary for this framework, and defer further discussion after the proof of (3) in Sec. C. Many SOC-based neural samplers (Zhang & Chen, 2022; Vargas et al., 2023) solve (3) or equivalent problem (Richter & Berner, 2024; Zhu et al., 2025). In the following paragraphs, we provide continuous and discrete instantiations of the optimization problem (3).

**SOC Approaches for Continuous Diffusion Samplers**     Consider the following controlled stochastic differential equation (SDE) on the state space $\mathcal{X} = \mathbb{R}^d$:

$$\mathrm{d}X_t = (b_t(X_t) + \sigma_t u_t^\theta(X_t))\mathrm{d}t + \sigma_t \mathrm{d}W_t, \quad X_0 \sim \mu, \tag{4}$$

where $b : [0, T] \times \mathcal{X} \to \mathcal{X}$ and $\sigma : [0, T] \to \mathbb{R}$ are the base drift and noise schedule, respectively, and $W$ is the Brownian motion (BM). Let $\mathbb{P}^\theta$ be the path measure induced by (4). The reference path measure $\mathbb{P}^{\mathrm{ref}}$ is the path measure induced by (4) with $u^\theta \equiv 0$. By applying Girsanov's theorem (Särkkä & Solin, 2019), the optimization problem (3) can be reduced to the following SOC problem:

$$\min_\theta \mathbb{E}_{X \sim \mathbb{P}^\theta} \left[ \int_0^T \frac{1}{2} \|u_t^\theta(X_t)\|^2 \mathrm{d}t - r(X_T) \right], \tag{5}$$

Under the memoryless condition (2), the solution of (5) then yields a controlled process whose terminal marginal equals $\pi$. Such memoryless references can be constructed by suitable choices of $(b, \sigma, \mu)$. For example, Path Integral Sampler (PIS, Zhang & Chen (2022) and Adjoint Sampler (AS, Havens et al. (2025)) use $\mu = \delta_0$ and $b_t \equiv 0$. Denoising Diffusion Sampler (DDS) (Vargas et al., 2023) takes $b_t(x) = -\frac{\alpha_t}{2}x$, $\sigma_t = \sqrt{\alpha_t}$, and $\mu = \mathcal{N}(0, I)$ where $\exp\left(-\int_0^T \alpha_t \mathrm{d}t\right) \approx 0$. This scheduling choice is the same as Variance Preserving SDE (VPSDE) in Song et al. (2021b). In these cases, the reference terminal law $\nu$ is a tractable Gaussian, so $r(\cdot)$ in (5) is available in closed form.

**SOC Approaches for Discrete Diffusion Samplers**     A counterpart of SDE in a discrete state space $\mathcal{X}$ is the continuous-time Markov chain (CTMC), which is a stochastic process $X = (X_t)_{t \in [0, T]}$ taking values in $\mathcal{X}$. The distribution of a CTMC is determined by its **generator** $Q = (Q_t)_{t \in [0, T]}$, defined as $Q_t(x, y) := \lim_{h \to 0} \frac{1}{h}(\Pr(X_{t+h} = y | X_t = x) - 1_{x=y})$ for $x, y \in \mathcal{X}$.

Existing SOC approaches for training discrete neural samplers mainly rely on masked discrete diffusion (Zhu et al., 2025). We suppose the target distribution is supported on $\mathcal{X}_0 = \{1, 2, ..., N\}^d$ and let the mask-augmented state space be $\mathcal{X} = \{1, 2, ..., N, \mathbf{M}\}^d$. The typical choice of the reference path measure $\mathbb{P}^{\mathrm{ref}}$ has a generator $Q^{\mathrm{ref}}$ defined by $Q_t^{\mathrm{ref}}(x, x^{i \leftarrow n}) = \frac{\gamma(t)}{N} 1_{x^i = \mathbf{M}, n \neq \mathbf{M}}$, where $\gamma : [0, T] \to \mathbb{R}_+$ can be any function satisfying $\int_0^T \gamma(t) \mathrm{d}t = \infty$. By construction, $\mathbb{P}_0^{\mathrm{ref}} = \mu$ is the delta distribution on the fully masked sequence, denoted $p_{\mathrm{mask}}$ (which guarantees memorylessness), and $\mathbb{P}_T^{\mathrm{ref}} = \nu$ is the uniform distribution over $\mathcal{X}_0$, denoted $p_{\mathrm{unif}}$. SOC-based mask discrete diffusion neural samplers (Zhu et al., 2025) solve the following problem akin to (5):

$$\min_\theta \mathbb{E}_{X \sim \mathbb{P}^\theta} \left[ \int_0^T \sum_{y \neq X_t} \left( Q_t^\theta \log \frac{Q_t^\theta}{Q_t^{\mathrm{ref}}} - Q_t^\theta + Q_t^{\mathrm{ref}} \right)(X_t, y)\mathrm{d}t - r(X_T) \right],$$

$$\text{s.t. } X = (X_t)_{t \in [0, T]} \text{ is a CTMC on } \mathcal{X} \text{ with generator } Q^\theta, \; X_0 \sim p_{\mathrm{mask}},$$

where $r$ has the same definition as in (5). The generator $Q^\theta$ is parameterized by $Q_t^\theta(x, x^{i \leftarrow n}) = \gamma(t)s_\theta(x)_{i,n} 1_{x^i = \mathbf{M}, n \neq \mathbf{M}}$ where $s_\theta(x)$ is a $d \times N$ matrix whose each row is a probability vector.

## 2.2 WEIGHTED CROSS ENTROPY SAMPLERS

**Cross-entropy-based Sampler**     The optimization problem (3) can be viewed as minimizing the **relative-entropy (RE)** $\mathrm{KL}(\mathbb{P}^\theta \| \mathbb{P}^*)$. Another popular choice of the loss functional $\mathcal{F}$ is the **cross-entropy (CE)** loss $\mathrm{KL}(\mathbb{P}^* \| \mathbb{P}^\theta) = \mathbb{E}_{\mathbb{P}^*} \log \frac{\mathrm{d}\mathbb{P}^*}{\mathrm{d}\mathbb{P}^\theta}$. The CE loss requires *importance sampling* to evaluate, as we have no access to the ground-truth path measure $\mathbb{P}^*$. A common choice for the

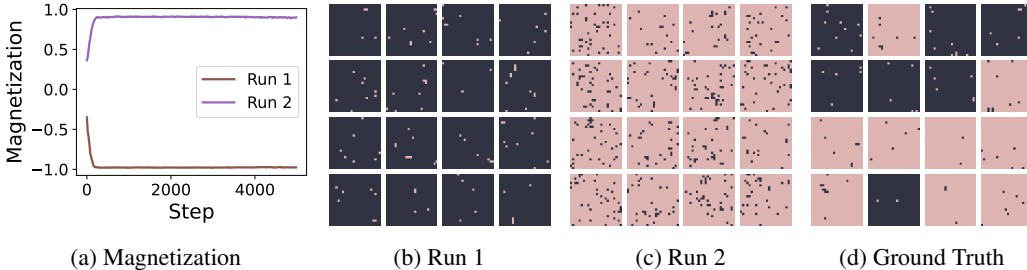

(a) Magnetization      (b) Run 1      (c) Run 2      (d) Ground Truth

Figure 1: An example of mode collapse in WDCE-based method where the target distribution is an Ising model under low temperature with two runs. (a) The magnetization (i.e., average spin) of the generated samples during training. The ground-truth value is $0$. We plot two independent runs, namely Run 1 and Run 2. (b, c) Visualization of the generated samples at the final training step for Run 1 and Run 2. (d) Visualization of the ground-truth samples.

proposal path measure is the *detached* path measure[3] $\mathbb{P}^{\bar{\theta}}$ for faster convergence. In this case, the CE loss can be expressed as follows:

$$\mathrm{KL}(\mathbb{P}^*\|\mathbb{P}^\theta) = \mathbb{E}_{\mathbb{P}^{\bar{\theta}}}\frac{\mathrm{d}\mathbb{P}^*}{\mathrm{d}\mathbb{P}^{\bar{\theta}}}\log\frac{\mathrm{d}\mathbb{P}^*}{\mathrm{d}\mathbb{P}^\theta} = \mathbb{E}_{X\sim\mathbb{P}^{\bar{\theta}}}\left(\frac{1}{Z}\frac{\mathrm{d}\mathbb{P}^{\mathrm{ref}}}{\mathrm{d}\mathbb{P}^{\bar{\theta}}}(X)\mathrm{e}^{r(X_T)}\right)\log\frac{\mathrm{d}\mathbb{P}^{\mathrm{ref}}}{\mathrm{d}\mathbb{P}^\theta}(X) + C, \quad (6)$$

where $C$ is a term independent of $\theta$.

**Weighted Denoising Cross-entropy (WDCE) Samplers** A drawback of the RE and CE losses is that they need to store the whole path of $X$ for computing loss. A more memory-efficient alternative is the weighted denoising cross-entropy (WDCE) loss which replaces negative log likelihood term in (6) with a *denoising score (or bridge) matching* objective (Phillips et al., 2024; Zhu et al., 2025). This objective is theoretically equivalent to the CE formulation and therefore implements the same reverse-KL objective up to an additive constant. Practically, WDCE allows us to retain only the terminal samples $X_T$ from rollouts $X \sim \mathbb{P}^\theta$. The pathwise weight $w(X) \propto \frac{\mathrm{d}\mathbb{P}^*}{\mathrm{d}\mathbb{P}^\theta}(X)$ can be accumulated online during simulation, so training proceeds from storing pairs $(X_T, w(X))$, without storing full trajectories. See Sec. 4 and Sec. C.2 for a detailed CE and WDCE formulation and its instantiation.

# 3   PROXIMAL DIFFUSION NEURAL SAMPLER: A GENERAL FRAMEWORK

## 3.1   TACKLING MODE COLLAPSE OF SOC-BASED SAMPLERS

Although SOC-based neural samplers are theoretically grounded, these methods suffer from **mode collapse** in the early stages of training, when $\mathbb{P}^\theta$ is far from the target $\mathbb{P}^*$. These samplers attempt to optimize a given SOC problem using the set of trajectories generated by the current model $\mathbb{P}^\theta$. When the distributional mismatch is large, only a small subset of trajectories carry a meaningful signal (e.g., high likelihood or high importance weight under $\mathbb{P}^*$), causing the objective to focus on this narrow subset of the trajectories. As a result, the losses are dominated by a few high-weight paths, producing unstable updates and poor coverage of the target. Moreover, due to the lack of exploration of the entire state space, these SOC-based methods tend to fit the modes that the existing trajectories have already reached, and reinforce this memory without attending to the rest of the possibilities, potentially leading to mode collapse.

To illustrate this effect, we train a WDCE sampler (Zhu et al., 2025) on a discrete benchmark: $24 \times 24$ lattice Ising model under low temperature $\beta = 0.6$, where $E(x) = \sum_{i\sim j} x_i x_j$, $x \in \{\pm 1\}^{24\times 24}$, and $i \sim j$ means $i, j$ are adjacent. At this temperature, the target distribution $\pi$ is bimodal, concentrating around the two ferromagnetic states (all-positive spins or all-negative spins) separated by a large energy barrier. As shown in the training dynamics in Fig. 1, WDCE rapidly collapses into a single mode and subsequently reinforces this mode through self-generated samples.

---

[3]Sample the path from $\mathbb{P}^\theta$ and detach it from the computational graph.

### 3.2 PROXIMAL DIFFUSION NEURAL SAMPLER

We develop the **Proximal Diffusion Neural Sampler (PDNS)** to mitigate the mode collapse observed in Sec. 3.1. Rather than directly solving the global SOC objective in (3), PDNS proceeds by incrementally solving *local* sub-problems that progressively move the current model toward the target distribution. Concretely, we apply **proximal point method** (Parikh & Boyd, 2014) in the space of path measures, progressively driving $\mathbb{P}^\theta$ toward $\mathbb{P}^*$ step-by-step while preserving mode coverage.

**Proximal Point Method for Diffusion Samplers** To transform (3) into a local sub-problem, we add a proximity term by a KL divergence from the previous iterate $\mathbb{P}^{\theta_{k-1}}$. Formally, we define the proximal point sub-problem with *proximal step size* $\eta_k \in (0, \infty]$ as follows:

$$\mathbb{P}^{\theta_k^*} = \underset{\mathbb{P}^\theta}{\arg\min} \left( -\mathbb{E}_{\mathbb{P}^\theta(X)} \, r(X_T) + \mathrm{KL}(\mathbb{P}^\theta \| \mathbb{P}^{\mathrm{ref}}) + \frac{1}{\eta_k} \mathrm{KL}(\mathbb{P}^\theta \| \mathbb{P}^{\theta_{k-1}}) \right). \tag{7}$$

Rather than solving the difficult SOC problem (3) (corresponding to infinite step size $\eta_k$), the problem (7) searches for the minimizer of the KL-penalized optimization problem. This KL-penalization term ensures that $\mathbb{P}^{\theta_k^*}$ remains close to the previous iterate $\mathbb{P}^{\theta_{k-1}}$, thereby resulting more robust updates and simplifying the optimization due to the dominant regularization effect. In practice, for a given path, the proximal objective effectively tempers the importance weights $\frac{d\mathbb{P}^{\theta_k^*}}{d\mathbb{P}^{\theta_{k-1}}}$ compared to that of non-proximal counterpart $\frac{d\mathbb{P}^*}{d\mathbb{P}^{\theta_{k-1}}}$, reducing the dominance of a single mode, thereby mitigating mode collapse. Theoretically, one can show that $\mathbb{P}^{\theta_k^*} \to \mathbb{P}^*$ as $k \to \infty$. Moreover, when $\mathbb{P}^{\theta_0^*} = \mathbb{P}^{\mathrm{ref}}$, the optimal $\mathbb{P}^{\theta_k^*}$ can be written as a geometric interpolation between $\mathbb{P}^{\mathrm{ref}}$ and $\mathbb{P}^*$. We summarize in the following result, with proof in Sec. C.

**Proposition 3.1.** *1. The optimal solution of* (7) *is*

$$\mathbb{P}^{\theta_k^*} \propto \left( \mathbb{P}^{\theta_{k-1}} \right)^{\frac{1}{\eta_k + 1}} \left( \mathbb{P}^* \right)^{\frac{\eta_k}{\eta_k + 1}} \qquad \Longleftrightarrow \qquad \frac{d\mathbb{P}^{\theta_k^*}}{d\mathbb{P}^{\theta_{k-1}}} \propto \left( \frac{d\mathbb{P}^*}{d\mathbb{P}^{\theta_{k-1}}} \right)^{\frac{\eta_k}{\eta_k + 1}}. \tag{8}$$

*2. Assume for all $k \geq 1$, the subproblems are solved to optimality and let $\mathbb{P}^{\theta_0} \leftarrow \mathbb{P}^{\mathrm{ref}}$. Denote $\mathbb{P}^k$ as the corresponding path measure $\mathbb{P}^{\theta_k^*}$, which satisfies $\mathbb{P}^k \propto \left( \mathbb{P}^{k-1} \right)^{\frac{1}{\eta_k + 1}} \left( \mathbb{P}^* \right)^{\frac{\eta_k}{\eta_k + 1}}$. We thus have*

$$\mathbb{P}^k \propto \left( \mathbb{P}^{\mathrm{ref}} \right)^{\lambda_k} \left( \mathbb{P}^* \right)^{1 - \lambda_k}, \quad \text{where } \lambda_k := \prod_{i=1}^{k} \frac{1}{\eta_i + 1}. \tag{9}$$

*This implies $\mathbb{P}^k$ converges to $\mathbb{P}^*$ if $\lambda_k \to 0$. Moreover,*

$$\mathbb{P}^k(X) \propto \mathbb{P}^{\mathrm{ref}}(X) e^{(1 - \lambda_k) r(X_T)} \propto \mathbb{P}^{\mathrm{ref}}(X) \frac{\pi^{1 - \lambda_k} \nu^{\lambda_k}}{\nu}(X_T), \tag{10}$$

*i.e., the terminal distribution of $\mathbb{P}^k$ is $\mathbb{P}_T^k \propto \pi^{1 - \lambda_k} \nu^{\lambda_k}$, the geometric interpolation between $\nu$ and $\pi$.*

In other words, iteratively solving the local optimization problems yields a sequence of path measures $\{\mathbb{P}^{\theta_k}\}$ or $\{\mathbb{P}^k\}$ that converges to $\mathbb{P}^*$. For simplicity, we use $\mathbb{P}^{k^*}$ to denote the optimal solution to the $k$-th proximal step, which can be chosen as either $\mathbb{P}^{\theta_k^*}$ (8) or $\mathbb{P}^k$ (9). We summarize the meanings of the notations of path measures in Tab. 1, and present the overall PDNS framework in Alg. 1.

Here, the *proximal step size* $\eta_k$ is a crucial hyperparameter: smaller $\eta_k$ increases the relative weight of the KL regularizer, resulting in more conservative updates that better preserve mode coverage. Instead, it has

Table 1: Explanation of the path measures (P.M.).

| | |
|---|---|
| $\mathbb{P}^\theta$ | General notation for P.M. of a neural sampler parameterized by $\theta$ |
| $\mathbb{P}^{\mathrm{ref}}, \mathbb{P}^*$ | Reference and optimal P.M.s defined in (3) |
| $\mathbb{P}^{\theta_{k-1}}$ | P.M. of the neural sampler at the end of the $(k-1)$-th proximal step |
| $\mathbb{P}^{\theta_k^*}$ | Optimal P.M. for $k$-th proximal step, starting from $\mathbb{P}^{\theta_{k-1}}$, i.e. (8) |
| $\mathbb{P}^k$ | Exact solution P.M. for $k$-th proximal step, i.e. (9) and (10) |
| $\mathbb{P}^{k^*}$ | either $\mathbb{P}^{\theta_k^*}$ or $\mathbb{P}^k$, training objective at the $k$-th proximal step |

slower convergence to optimum. Larger $\eta_k$ weakens the regularization, enabling faster convergence, but risking mode collapse. Note that it is the step size $\eta_k$ affects performance due to this trade-off between convergence and mode coverage. We further discuss the possible proximal step size scheduler in the last paragraph of Sec. 4.

**Proximal CE for Diffusion Samplers** By reversing the arguments in the KL divergence of (3), we have obtained a cross-entropy counterpart (6). Likewise, we can obtain a cross-entropy counterpart to the proximal subproblem (7). Given the current iterate $\mathbb{P}^\theta$, we define the **proximal cross-entropy** update as $\min_{\mathbb{P}^\theta} \mathrm{KL}\left( \mathbb{P}^{\theta_k^*} \| \mathbb{P}^\theta \right)$ or $\min_{\mathbb{P}^\theta} \mathrm{KL}\left( \mathbb{P}^k \| \mathbb{P}^\theta \right)$. By leveraging the right-hand side of (3) and (8), the former object reduces to weighted negative log-likelihoods under

---

**Algorithm 1** Proximal Diffusion Neural Sampler (PDNS)

---

**Require:** Model parameters $\theta$, target distribution $\pi \propto e^{-\beta E}$, schedule of step sizes $\{\eta_k\}_{k \geq 1}$, reference path measure $\mathbb{P}^{\text{ref}}$, initial path measure $\mathbb{P}^{\theta_0}$, loss function $\mathcal{F}$.

1: **for** $k$ **in** $1, 2, \dots$ **do**
2:     Set $\mathbb{P}^{k^*} \in \{\mathbb{P}^{\theta_k^*}, \mathbb{P}^k\}$, the optimal solution to the current subproblem, following (8) or (10).
3:     **for** step **in** $1, 2, \dots$ **do**
4:         Compute the loss $\mathcal{F}(\mathbb{P}^\theta; \mathbb{P}^{k^*})$ using samples from $\mathbb{P}^{\theta_{k-1}}$ and update $\theta$.
5:     **end for**
6:     Set $\mathbb{P}^{\theta_k} \leftarrow \mathbb{P}^\theta$.
7: **end for**

---

the previous iterate, up to positive multiplicative factors and additive constants independent of $\mathbb{P}^\theta$:

$$
\begin{aligned}
\text{KL}(\mathbb{P}^{\theta_k^*} \| \mathbb{P}^\theta) &= \mathop{\mathbb{E}}_{X \sim \mathbb{P}^{\bar{\theta}_{k-1}}} \left[ \frac{d\mathbb{P}^{\theta_k^*}}{d\mathbb{P}^{\bar{\theta}_{k-1}}}(X) \log \frac{d\mathbb{P}^{\theta_k^*}}{d\mathbb{P}^\theta}(X) \right] \propto \mathop{\mathbb{E}}_{X \sim \mathbb{P}^{\bar{\theta}_{k-1}}} \left[ \left( \frac{d\mathbb{P}^*}{d\mathbb{P}^{\bar{\theta}_{k-1}}}(X) \right)^{\frac{\eta_k}{\eta_k+1}} \log \frac{d\mathbb{P}^{\theta_k^*}}{d\mathbb{P}^\theta}(X) \right] \\
&\propto -\mathop{\mathbb{E}}_{X \sim \mathbb{P}^{\bar{\theta}_{k-1}}} \left( e^{r(X_T)} \frac{d\mathbb{P}^{\text{ref}}}{d\mathbb{P}^{\bar{\theta}_{k-1}}}(X) \right)^{\frac{\eta_k}{\eta_k+1}} \log \mathbb{P}^\theta(X) + C,
\end{aligned} \tag{11}
$$

Furthermore, leveraging (10), we obtain

$$
\text{KL}(\mathbb{P}^k \| \mathbb{P}^\theta) \propto -\mathop{\mathbb{E}}_{X \sim \mathbb{P}^{k-1}} \left( e^{(1-\lambda_k)r(X_T)} \frac{d\mathbb{P}^{\text{ref}}}{d\mathbb{P}^{k-1}}(X) \right) \log \mathbb{P}^\theta(X) + C, \tag{12}
$$

up to some multiplicative normalizing constant and some irrelevant constant $C$. Note that both objectives (11) and (12) are reduced to a weighted log-likelihood objective.

**Proximal WDCE for Diffusion Samplers** As described in Sec. 2.2, the CE loss (6) can be approximated by the WDCE loss by simply replacing the negative log-likelihood of the CE term into the denoising score matching loss. Similarly, we can also further analyze the proximal CE losses (11)-(12) and derive a **proximal weighted denoising cross-entropy (proximal WDCE)** losses, denoted as $\min_\theta \mathcal{F}(\mathbb{P}^\theta; \mathbb{P}^{\theta_k^*})$ or $\min_\theta \mathcal{F}(\mathbb{P}^\theta; \mathbb{P}^k)$, depending on which local optimum we are targeting. In the next section, we provide explicit forms of the proximal WDCE objective $\mathcal{F}$ in both continuous and discrete settings.

# 4 PROXIMAL DIFFUSION NEURAL SAMPLER IN PRACTICE: CONTINUOUS AND DISCRETE CASES

Among various PDNS instantiations, we focus on a proximal WDCE variant of PDNS, selected for its practical effectiveness: it avoids storing entire trajectories and enjoys efficiency of (discrete) score matching objectives (Zhu et al., 2025). Now, we derive explicit forms of the proximal WDCE training objective $\mathcal{F}$ in Alg. 1 for both continuous and discrete diffusion settings. Furthermore, we also provide principled guidelines for selecting the proximal step sizes $\{\eta_k\}_{k=1}^\infty$. Complete algorithms for proximal WDCE and scheduler policy are provided in Sec. B and additional implementation details are provided in Secs. D and E.

**Proximal WDCE for Continuous Diffusion Samplers** By applying Diffusion Schödinger Bridge Matching (DSBM) (Shi et al., 2024), the proximal WDCE objective is written as follows:

$$
\begin{aligned}
\text{KL}(\mathbb{P}^{k^*} \| \mathbb{P}^\theta) &= \mathop{\mathbb{E}}_{t \sim \text{Unif}(0,T),\ X \sim \mathbb{P}^{k^*}} \left[ \frac{1}{2} \| u_t^\theta(X_t) - \sigma_t \nabla \log \mathbb{P}^{\text{ref}}_{T|t}(X_T | X_t) \|^2 \right] \\
&= \mathop{\mathbb{E}}_{t \sim \text{Unif}(0,T),\ X \sim \mathbb{P}^{\bar{\theta}_{k-1}}} \left[ \frac{d\mathbb{P}^{k^*}}{d\mathbb{P}^{\bar{\theta}_{k-1}}}(X) \frac{1}{2} \| u_t^\theta(X_t) - \sigma_t \nabla \log \mathbb{P}^{\text{ref}}_{T|t}(X_T | X_t) \|^2 \right].
\end{aligned} \tag{13}
$$

Here, the first line is the original DSBM loss with target path measure $\mathbb{P}^{k^*}$, and the second line we use importance sampling to rewrite the expectation under the detached path measure $\mathbb{P}^{\bar{\theta}_{k-1}}$ obtained from the previous iteration. We sample $N$ trajectories $\{X^{(i)}\}_{i=1}^N \sim \mathbb{P}^{\bar{\theta}_{k-1}}$ and compute the corresponding weights (RN derivative) $\left\{ w^{(i)} = d\mathbb{P}^{k^*} / d\mathbb{P}^{\bar{\theta}_{k-1}}(X^{(i)}) \right\}_{i=1}^N$, which will be discussed

in the next paragraph. Then, we store pairs $(X_T^{(i)}, w^{(i)})$ for $i \in \{1, \ldots, N\}$ in a buffer $\mathcal{B}$. In the training iteration, we sample

$$(X_T, w) \sim \mathcal{B}, \quad X_0 \sim \mu, \quad t \sim \mathrm{Unif}(0, T),$$

and sample $X_t$ using a *reciprocal property* (Léonard et al., 2014), i.e., $\mathbb{P}_{t|0,T}^{\theta_{k-1}} = \mathbb{P}_{t|0,T}^{\mathrm{ref}}$, to compute the matching objective in the equation. The detailed discussion is provided in Sec. C.2.

We now introduce closed form to compute the weight (RN derivative) in (13). Let $u_t^{\bar{\theta}_{k-1}}(x)$ be the control that induces the path measure $\mathbb{P}^{\theta_{k-1}}$. Using the Girsanov theorem (Øksendal, 2003),

$$\frac{\mathrm{d}\mathbb{P}^{\mathrm{ref}}}{\mathrm{d}\mathbb{P}^{\bar{\theta}_{k-1}}}(X) = \exp\left(-\int_0^T \frac{1}{2}\|u_t^{\bar{\theta}_{k-1}}(X_t)\|^2 \mathrm{d}t + u_t^{\bar{\theta}_{k-1}}(X_t) \cdot \mathrm{d}W_t\right), \tag{14}$$

so the RN derivative $\frac{\mathrm{d}\mathbb{P}^{k^*}}{\mathrm{d}\mathbb{P}^{\bar{\theta}_{k-1}}}$ in (13), where $\mathbb{P}^{k^*} \in \{\mathbb{P}^{\theta_k^*}, \mathbb{P}^k\}$, can be computed as follows:

$$\frac{\mathrm{d}\mathbb{P}^{\theta_k^*}}{\mathrm{d}\mathbb{P}^{\bar{\theta}_{k-1}}}(X) \propto \left(\mathrm{e}^{r(X_T)}\frac{\mathrm{d}\mathbb{P}^{\mathrm{ref}}}{\mathrm{d}\mathbb{P}^{\bar{\theta}_{k-1}}}(X)\right)^{\frac{\eta_k}{\eta_k+1}} \quad \text{or} \quad \frac{\mathrm{d}\mathbb{P}^k}{\mathrm{d}\mathbb{P}^{\bar{\theta}_{k-1}}} \propto \mathrm{e}^{(1-\lambda_k)r(X_T)}\frac{\mathrm{d}\mathbb{P}^{\mathrm{ref}}}{\mathrm{d}\mathbb{P}^{\bar{\theta}_{k-1}}}(X). \tag{15}$$

The detailed discussions are provided in Sec. C.2.

**Proximal WDCE for Discrete Diffusion Samplers**     The WDCE objective, following Ou et al. (2025a); Zhu et al. (2025), can be written as

$$\mathrm{KL}(\mathbb{P}^{k^*}\|\mathbb{P}^\theta) = \mathbb{E}_{X\sim\mathbb{P}^{\bar{\theta}_{k-1}}} \frac{\mathrm{d}\mathbb{P}^{k^*}}{\mathrm{d}\mathbb{P}^{\bar{\theta}_{k-1}}}(X) \mathbb{E}_{\lambda\sim\mathrm{Unif}(0,1)}\left[\frac{1}{\lambda}\mathbb{E}_{\mu_\lambda(\widetilde{x}|X_T)}\sum_{d:\widetilde{x}^d=\mathbf{M}} -\log s_\theta(\widetilde{x})_{d,X_T^d}\right],$$

where $\mu_\lambda(\cdot|x)$ means independently masking each entry of $x$ with probability $\lambda$, and $s_\theta$ is the parameterization of $\mathbb{P}^\theta$ introduced in Sec. 2.1.

We adopt (15) to compute the weight $\frac{\mathrm{d}\mathbb{P}^{k^*}}{\mathrm{d}\mathbb{P}^{\bar{\theta}_{k-1}}}$ for discrete cases, where $\mathbb{P}^{k^*} \in \{\mathbb{P}^{\theta_k^*}, \mathbb{P}^k\}$. It suffices to compute the following RN derivative $\frac{\mathrm{d}\mathbb{P}^{\mathrm{ref}}}{\mathrm{d}\mathbb{P}^{\bar{\theta}_{k-1}}}$ to obtain the target weight. Through the Girsanov theorem of CTMCs (Ren et al., 2025a;b; Zhu et al., 2025), for any trajectory $X$, we could obtain the RN derivative as follows:

$$\frac{\mathrm{d}\mathbb{P}^{\mathrm{ref}}}{\mathrm{d}\mathbb{P}^{\bar{\theta}_{k-1}}}(X) = \exp\left(\sum_{t:X_{t_-}\neq X_t} \log \frac{1/N}{s_{\bar{\theta}_{k-1}}(X_{t_-})_{i(t),X_t^{i(t)}}}\right), \tag{16}$$

where we assume the jump at time $t$ happens at the $i(t)$-th entry, and the total number of jumps is the sequence length $d$.

**Design of Schedulers**     As discussed in Sec. 3.2, the design of the step sizes $\{\eta_k\}_{k\geq 1}$ is important for the PDNS framework, which we refer to as the **scheduler** because it governs the trade-off between convergence speed and stability. A larger $\eta_k$ enables faster convergence but risks difficulty from larger distributional gaps, whereas smaller $\eta_k$ yields conservative yet more stable progress.

The simplest design is to choose a **predefined** schedule for $\eta_k$ or $\lambda_k$, leveraging the relationship in (9) and making sure $\lambda_k \to 0$. Unlike directly training the model to fit the target distribution $\pi$, the knowledge of the optimal solution of each subproblem (Prop. 3.1) in PDNS provides information on how the samples fit the local target distributions. A demonstration can be found in Fig. 12.

We also propose an **adaptive** strategy that selects $\eta_k$ automatically based on the current state of the model. The key idea is to ensure that the next target $\mathbb{P}^{k^*} \in \{\mathbb{P}^{\theta_k^*}, \mathbb{P}^k\}$ is not too far from the current one $\mathbb{P}^{\theta_{k-1}}$, e.g., through controlling $\widehat{\mathrm{KL}}(\mathbb{P}^{\theta_{k-1}}\|\mathbb{P}^{k^*}) \leq \epsilon$. Detailed description of the criteria is deferred to Sec. B, and we refer readers to Secs. D.3 and E.3 for ablation study on the schedulers.

## 5   EXPERIMENTS

We now demonstrate the effectiveness of our proposed PDNS framework on learning continuous and discrete target distributions. All implementation details and additional experimental results can be found in Secs. D and E.

Table 2: Evaluation on the synthetic energy functions. Following Chen et al. (2025); Choi et al. (2025), we report Sinkorn distance (↓) and MMD (↓). The **best** results and the second best results are highlighted.

| Method | MW54 ($d=5$) Sinkhorn | Funnel ($d=10$) MMD | Funnel ($d=10$) Sinkhorn | GMM40 ($d=50$) Sinkhorn | MoS ($d=50$) MMD | MoS ($d=50$) Sinkhorn |
|---|---|---|---|---|---|---|
| SMC (Del Moral et al., 2006) | $20.71_{\pm 5.33}$ | - | $149.35_{\pm 4.73}$ | $46370.34_{\pm 137.79}$ | - | $3297.28_{\pm 2184.54}$ |
| SMC-ESS (Buchholz et al., 2021) | $1.11_{\pm 0.15}$ | - | $\mathbf{117.48}_{\pm 9.70}$ | $24240.68_{\pm 50.52}$ | - | $1477.04_{\pm 133.80}$ |
| CRAFT (Arbel et al., 2021) | $11.47_{\pm 0.90}$ | $0.115_{\pm 0.003}$ | $134.34_{\pm 0.66}$ | $28960.70_{\pm 354.89}$ | $0.257_{\pm 0.024}$ | $1918.14_{\pm 108.22}$ |
| DDS (Vargas et al., 2023) | $0.63_{\pm 0.24}$ | $0.172_{\pm 0.031}$ | $142.89_{\pm 9.55}$ | $5435.18_{\pm 172.20}$ | $0.131_{\pm 0.001}$ | $2154.88_{\pm 3.861}$ |
| PIS (Zhang & Chen, 2022) | $0.42_{\pm 0.01}$ | - | - | $10405.75_{\pm 69.41}$ | $0.218_{\pm 0.007}$ | $2113.17_{\pm 31.17}$ |
| AS (Havens et al., 2025) | $0.32_{\pm 0.06}$ | - | - | $18984.21_{\pm 62.12}$ | $0.210_{\pm 0.004}$ | $2178.60_{\pm 54.82}$ |
| CMCD-KL (Vargas et al., 2024) | $0.57_{\pm 0.05}$ | $0.095_{\pm 0.003}$ | $513.33_{\pm 192.4}$ | $22132.28_{\pm 595.18}$ | - | $1848.89_{\pm 532.56}$ |
| CMCD-LV (Vargas et al., 2024) | $0.51_{\pm 0.08}$ | - | $139.07_{\pm 9.35}$ | $4258.57_{\pm 737.15}$ | - | $1945.71_{\pm 48.79}$ |
| SCLD (Chen et al., 2025) | $0.44_{\pm 0.06}$ | - | $134.23_{\pm 8.39}$ | $3787.73_{\pm 249.75}$ | - | $656.10_{\pm 88.97}$ |
| NAAS (Choi et al., 2025) | $0.10_{\pm 0.0075}$ | $\mathbf{0.076}_{\pm 0.004}$ | $132.30_{\pm 5.87}$ | $496.48_{\pm 27.08}$ | $0.113_{\pm 0.070}$ | $394.55_{\pm 29.35}$ |
| PDNS | $\mathbf{0.08}_{\pm 0.01}$ | $0.086_{\pm 0.006}$ | $129.54_{\pm 4.15}$ | $\mathbf{327.83}_{\pm 54.68}$ | $\mathbf{0.059}_{\pm 0.017}$ | $\mathbf{353.05}_{\pm 68.63}$ |

Table 3: Evaluation on the particle-based energy functions. Following Havens et al. (2025); Liu et al. (2025), we report the Wasserstein-2 distances w.r.t samples, $\mathcal{W}_2(\downarrow)$, and energies, $E(\cdot)\mathcal{W}_2(\downarrow)$. The **best** results and the second best results are highlighted.

| Method | DW-4 ($d=8$) $\mathcal{W}_2\downarrow$ | DW-4 ($d=8$) $E(\cdot)\mathcal{W}_2\downarrow$ | LJ-13 ($d=39$) $\mathcal{W}_2\downarrow$ | LJ-13 ($d=39$) $E(\cdot)\mathcal{W}_2\downarrow$ | LJ-55 ($d=165$) $\mathcal{W}_2\downarrow$ | LJ-55 ($d=165$) $E(\cdot)\mathcal{W}_2\downarrow$ |
|---|---|---|---|---|---|---|
| PDDS (Phillips et al., 2024) | $0.92_{\pm 0.08}$ | $0.58_{\pm 0.25}$ | $4.66_{\pm 0.87}$ | $56.01_{\pm 10.80}$ | — | — |
| SCLD (Chen et al., 2025) | $1.30_{\pm 0.64}$ | $0.40_{\pm 0.19}$ | $2.93_{\pm 0.19}$ | $27.98_{\pm 1.26}$ | — | — |
| PIS (Zhang & Chen, 2022) | $0.68_{\pm 0.28}$ | $0.65_{\pm 0.25}$ | $1.93_{\pm 0.07}$ | $18.02_{\pm 1.12}$ | $4.79_{\pm 0.45}$ | $228.70_{\pm 131.27}$ |
| DDS (Vargas et al., 2023) | $0.92_{\pm 0.11}$ | $0.90_{\pm 0.37}$ | $1.99_{\pm 0.13}$ | $24.61_{\pm 8.99}$ | $4.60_{\pm 0.09}$ | $173.09_{\pm 18.01}$ |
| LV-PIS (Richter & Berner, 2024) | $1.04_{\pm 0.29}$ | $1.89_{\pm 0.89}$ | — | — | — | — |
| iDEM (Akhound-Sadegh et al., 2024) | $0.70_{\pm 0.06}$ | $0.55_{\pm 0.14}$ | $1.61_{\pm 0.01}$ | $30.78_{\pm 24.46}$ | $4.69_{\pm 1.52}$ | $93.53_{\pm 16.31}$ |
| AS (Havens et al., 2025) | $0.62_{\pm 0.06}$ | $0.55_{\pm 0.12}$ | $1.67_{\pm 0.01}$ | $2.40_{\pm 1.25}$ | $4.50_{\pm 0.05}$ | $58.04_{\pm 20.98}$ |
| ASBS (Liu et al., 2025) | $\mathbf{0.38}_{\pm 0.05}$ | $\mathbf{0.19}_{\pm 0.03}$ | $1.59_{\pm 0.00}$ | $1.28_{\pm 0.22}$ | $4.00_{\pm 0.03}$ | $27.69_{\pm 3.86}$ |
| PDNS | $0.51_{\pm 0.04}$ | $0.21_{\pm 0.03}$ | $\mathbf{1.57}_{\pm 0.01}$ | $\mathbf{1.01}_{\pm 0.18}$ | $\mathbf{3.95}_{\pm 0.01}$ | $\mathbf{21.97}_{\pm 3.14}$ |

## 5.1 LEARNING CONTINUOUS TARGET DISTRIBUTIONS

**Overview**     We evaluate our model on the following continuous target benchmarks:

- *Continuous synthetic energy functions*     Following (Choi et al., 2025), we consider four different multi-modal synthetic benchmarks; the 32-well many-well potential (MW-54, 5D), Funnel (5D), a 40-component Gaussian mixture (GMM40, 50D), and a mixture of Student's $t$ distribution (MoS, 50D) are considered. We evaluate with two metrics; the maximum mean discrepancy (MMD) (Akhound-Sadegh et al., 2024) and the Sinkhorn distance (Sinkhorn) with a small entropic regularization ($10^{-3}$) (Chen et al., 2025).

- *Particle potentials*     We consider classical continuous potentials where the energy $E(x)$ is analytic and depends on pairwise distances in an $n$-particle system. Following Akhound-Sadegh et al. (2024), we consider DW-4 (2D, 4 particles), LJ-13 (3D, 13 particles), and LJ-55 (3D, 55 particles). We use the equivariant 2-Wasserstein distance and 2-Wasserstein distance between the ground-truth and the generated samples, following (Akhound-Sadegh et al., 2024; Liu et al., 2025). For the ground-truth samples, we use MCMC samples from Klein et al. (2023).

- *Alanine Dipeptide (AD)*     The AD system contains 22 atoms in three-dimensional space. Following prior work (Midgley et al., 2023), we simulate the molecule using the OpenMM package (Eastman et al., 2017) and parameterize its geometry in terms of internal coordinates, resulting in a 60-dimensional representation. For evaluation, we compare the distribution of generated samples against the reference distribution using the KL divergence computed from $10^4$ samples. Our evaluation metric is KL divergence (DKL) between generated and reference samples for the backbone angles $\phi$, $\psi$ and the methyl rotation angles $\gamma_1$, $\gamma_2$, $\gamma_3$. In addition to these quantitative metrics, we also visualize the energy histogram, torsion plot, and Ramachandran plot for both generated and reference samples to assess structural fidelity. We use $10^6$ samples for torsion plot and Ramachandran plot.

Table 4: Comparison between diffusion samplers on sampling the molecular Boltzmann distribution of the alanine dipeptide. We report the KL divergence KL for the 1D marginal across five torsion angles.

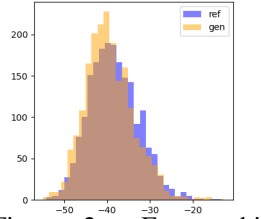

|  | KL on each torsion's marginal ↓ | | | | |
|---|---|---|---|---|---|
| Method | $\phi$ | $\psi$ | $\gamma_1$ | $\gamma_2$ | $\gamma_3$ |
| PIS (Zhang & Chen, 2022) | 0.05 | 0.38 | 5.61 | 4.49 | 4.60 |
| DDS (Vargas et al., 2023) | 0.03 | 0.16 | 2.44 | 0.03 | 0.03 |
| AS (Havens et al., 2025) | 0.09 | 0.04 | 0.17 | 0.56 | 0.51 |
| ASBS (Liu et al., 2025) | 0.02 | 0.01 | 0.03 | 0.02 | 0.02 |
| PDNS (Ours) | 0.02 | 0.04 | 0.03 | 0.02 | 0.02 |

Figure 2: Energy histograms for AD.

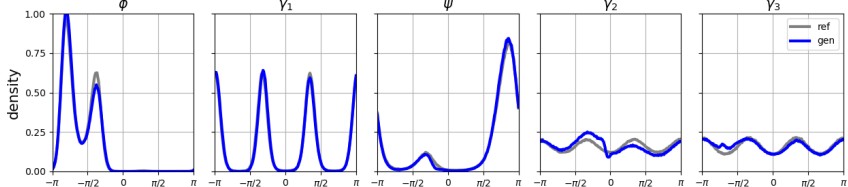

Figure 3: Comparison of torsions between PDNS and reference.

Detailed benchmark descriptions, experimental settings, and evaluation metrics are provided in Sec. D. In Sec. D.3, we also compare the WDCE baseline, the main baseline of our models to our *proximal* WDCE. Moreover, in Secs. D.3 and D.4, we further present qualitative and quantitative results, including ablations over hyperparameters and schedulers, and sample visualizations.

**Main Results** Quantitative results are demonstrated in Tabs. 2 and 3. Among seven benchmarks, PDNS (with proximal WDCE) attains the best scores on five tasks, with the only exceptions being Funnel and DW4, where performance is comparable but slightly behind recent baselines. Notably, PDNS excels on the more challenging targets MoS (50D) with heavy tails and the LJ-13/LJ-55 particle systems with steep, rugged energy surfaces. We attribute these gains to the proximal step, which tempers trajectory weights and constrains each iteration to a local move in path space. This prevents the model from committing to a single model early, promotes gradual exploration across separated modes, and stabilizes optimization in high dimensions. As illustrated in Sec. D.3, removal of the proximal term induces mode collapse, whereas the PDNS maintains coverage. Finally, as shown in Tab. 4, Figs. 2 and 3, our method achieves strong performance on the alanine dipeptide task. Quantitatively, the KL divergences of the torsion angles are comparable to those reported by the current state-of-the-art method, ASBS (Liu et al., 2025), and the qualitative assessments including energy histograms and torsion plots, demonstrate that our model captures the main conformational structures of the system.

## 5.2 LEARNING DISCRETE TARGET DISTRIBUTIONS

**Overview** We evaluate our method on the following discrete sampling tasks:

- *Learning distributions from statistical physics* Following Holderrieth et al. (2025); Zhu et al. (2025), we consider the Ising and Potts models on a square lattice with $L$ sites per dimension, which are two classical models from statistical physics that exhibit phase transitions under different inverse temperatures (Onsager, 1944; Beffara & Duminil-Copin, 2012). To emphasize the advantage of our method in learning distributions with mode separation, we mainly consider temperatures around or below the critical one: for Ising model, we choose $L = 24$, and consider $\beta_{\text{critical}} = \log(1 + \sqrt{2})/2 = 0.4407$ and $\beta_{\text{low}} = 0.6$; for Potts model, we choose $L = 16$, number of states $q = 4$, and consider $\beta_{\text{critical}} = \log(1 + \sqrt{q}) = 1.0986$ and $\beta_{\text{low}} = 1.3$. We compare our method with LEAPS (Holderrieth et al., 2025) and the Metropolis-Hastings (MH) algorithm, and use the Swendsen-Wang (SW) algorithm (Swendsen & Wang, 1986; 1987) to generate ground-truth samples for evaluation. As the original WDCE loss cannot learn the correct target distribution (Fig. 1), we do not include its performance here.

- *Amortized sampling for combinatorial optimization* Recently, there is a trend of leveraging sampling methods for solving combinatorial optimization problems (Sun et al., 2022; Sanokowski et al., 2024; Ou et al., 2025b), by writing the loss function as the energy $E$ and sampling from the distribution $\pi_\beta \propto e^{-\beta E}$ under a relatively large inverse temperature (e.g., $\beta = 5$), so that $\pi_\beta$ concentrates on the

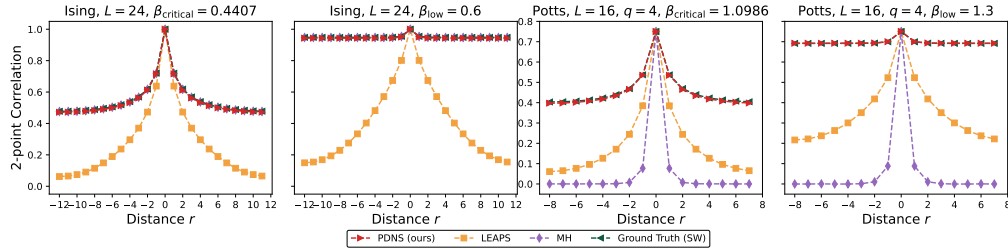

Figure 4: Average 2-point correlations in both vertical and horizontal directions of samples from learned Ising and Potts models at different inverse temperatures.

global minimum of $E$, thus improving the optimization accuracy. This low-temperature nature of the target distributions serves as a perfect example to demonstrate the effectiveness of PDNS.

**Main Results** For sampling from Ising and Potts models, we report quantitative results, including error in magnetization and 2-point correlation with respect to the ground-truth samples in Tab. 5, and visualize the learned average 2-point correlations at states differing at different distances $r$ in Fig. 4. We observe that our method significantly outperforms LEAPS. While the MH algorithm can achieve good accuracy in Ising model, its performance degrades significantly in Potts model due to the larger number of spin states, yet our method still achieves good performance. For combinatorial optimization, in Tab. 6, we use PNDS to solve the maximum cut problem and test the performance on two classical random graphs. We report the solution compared with the ground-truth value solved by Gurobi (Gurobi Optimization, LLC, 2025), showing PDNS's capability of reaching comparable performance. We defer detailed explanation of the distributions and settings to Sec. E.2.

Table 5: Results for learning Ising and Potts models at different inverse temperatures, the best result is highlighted. Mag. and Corr. represent absolute errors of magnetization and average 2-point correlations to ground-truth values given by the SW algorithm.

| Distribution | Ising model, $L = 24$, $J = 1$ | | | | | | Potts model, $L = 16$, $J = 1$, $q = 4$ | | | | | |
|---|---|---|---|---|---|---|---|---|---|---|---|---|
| Inv. Temp. | $\beta_{\text{critical}} = 0.4407$ | | | $\beta_{\text{low}} = 0.6$ | | | $\beta_{\text{critical}} = 1.0986$ | | | $\beta_{\text{low}} = 1.3$ | | |
| Metrics | Mag. ↓ | Corr. ↓ | ESS ↑ | Mag. ↓ | Corr. ↓ | ESS ↑ | Mag. ↓ | Corr. ↓ | ESS ↑ | Mag. ↓ | Corr. ↓ | ESS ↑ |
| PDNS (ours) | **1.2e − 2** | 5.6e − 3 | **0.903** | 9.0e − 3 | 4.7e − 3 | **0.950** | **5.2e − 3** | **4.6e − 3** | **0.948** | **8.4e − 4** | **6.1e − 4** | **0.978** |
| LEAPS | 5.9e − 2 | 2.8e − 1 | 0.020 | 3.0e − 2 | 5.5e − 1 | 0.001 | 3.2e − 1 | 2.6e − 1 | 0.112 | 3.6e − 1 | 3.5e − 1 | 0.021 |
| Baseline (MH) | 2.2e − 2 | **1.9e − 3** | / | **1.6e − 3** | **6.6e − 4** | / | 5.3e − 1 | 4.0e − 1 | / | 7.6e − 1 | 6.4e − 1 | / |

Table 6: Results for finding maximum cut of Barabási–Albert (BA) and Erdős–Rényi (ER) graphs by PNDS sampling. The evaluation metrics are maximum and average cut size, i.e., the maximum / average of the size of all generated samples for each graph, taken average over all graphs in the test set. Higher is better.

| Num. of Vtx. | $16 \sim 20$ | | $32 \sim 40$ | | $64 \sim 75$ | | $100 \sim 128$ | |
|---|---|---|---|---|---|---|---|---|
| Graph Type | BA | ER | BA | ER | BA | ER | BA | ER |
| Random | 0.689; 0.855 | 0.725; 0.873 | 0.685; 0.806 | 0.780; 0.861 | 0.678; 0.760 | 0.836; 0.879 | 0.675; 0.740 | 0.865; 0.894 |
| PDNS (ours) | 0.947; 0.991 | 0.962; 0.995 | 0.939; 0.973 | 0.966; 0.989 | 0.918; 0.947 | 0.970; 0.983 | 0.882; 0.915 | 0.973; 0.983 |

## 6 CONCLUSION

We propose Proximal Diffusion Neural Sampler (PDNS), a unified framework for SOC-based training of diffusion neural samplers that perform proximal point method on the path measure space. We instantiate PDNS with the proximal WDCE objective, and evaluate our method on various continuous and discrete target benchmarks where PDNS shows competitive performance. Our paper has two main limitations: we investigate a single instantiation of PDNS, and our experiments focus on controlled benchmarks rather than real-world, large scale applications. Potential directions include exploring SOC problem with memoryless reference dynamic $\mathbb{P}^{\text{ref}}$, other instantiations of PDNS (e.g., other proximal objectives or priors), and extending the framework for distributions on more general state spaces and neural samplers beyond the SOC formulation.

ACKNOWLEDGMENTS

WG thanks Zijing Ou and Sebastian Sanokowski for discussions on combinatorial optimization problems. We thank Petr Molodyk and Bo Yuan for discussions during the early stage of this work. WG, JC, and YC acknowledge supports from NSF Grants ECCS-1942523 and DMS-2206576. YZ and MT are thankful for partial supports by NSF Grants DMS-1847802, DMS-2513699, DOE Grants NA0004261, SC0026274, and Richard Duke Fellowship.

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

# A    RELATED WORKS

## A.1    CONTINUOUS DIFFUSION SAMPLERS

**SOC-based Diffusion Samplers**    Formulating diffusion-based sampling as a stochastic optimal control (SOC) problem has emerged as an active line of research. The Path Integral Sampler (PIS) (Zhang & Chen, 2022) and Denoising Diffusion Samplers (DDS) (Vargas et al., 2023) adopt *memoryless* reference dynamics and optimize the control either through direct gradient updates or adjoint-based training. LV-PIS (Richter & Berner, 2024) introduces a log-variance (LV) formulation, whose optimal solution coincides with the SOC optimal control. LV-based methods decouple the rollout path measure from the training objective, enabling flexible trajectory generation and avoiding the need to store full pathwise gradients or explicit adjoints; however, the loss still aggregates along entire trajectories. Adjoint Matching (AM) (Domingo-Enrich et al., 2025) takes an orthogonal approach by replacing full adjoint solves with a lean-adjoint estimator tailored to the SOC structure, thereby improving computational efficiency. Building on AM, the Adjoint Sampler (AS) (Havens et al., 2025) incorporates a reciprocity (optimality) condition on path measures to further enhance training efficacy. Two recent extensions broaden this framework: ASBS (Liu et al., 2025) generalizes to non-memoryless references inspired by the Schrödinger bridge problem, while NAAS (Choi et al., 2025) employs annealed reference dynamics to provide stronger guidance during learning.

**CE-based Diffusion Samplers**    Cross-entropy (CE) approaches replace forward-KL or relative-entropy training with a reverse-KL projection, typically of the form, which reduces to a weighted negative log-likelihood on trajectories. In the diffusion setting, this often appears as a denoising or score-matching style objective with importance weights to correct for mismatch between the rollout law and the target. Phillips et al. (2024) utilize weighted denoising cross-entropy (WDCE) objective derived from denoising score matching. Their sampler combines importance weighting with a diffusion-based learning loss to bridge from a simple reference to the target. Blessing et al. (2025) couple CE training with Adjoint Matching (AM), and interpret the resulting criterion within the stochastic optimal control matching (SOCM) viewpoint (Domingo-Enrich et al., 2024). They introduce a trust-region mechanism that can be cast as a proximal step or an explicit KL bound, yielding more stable CE iterations. This work is concurrent with ours. In contrast, PDNS utilize proximal algorithm with WDCE objective, which can be applied uniformly to continuous-time and discrete-time diffusions, and is designed to be memory-efficient while retaining the benefits of denoising or score-matching objectives.

**Sequential Monte Carlo and Annealed Diffusion Samplers**    A long line of work studies Markov chain Monte Carlo (MCMC) and Sequential Monte Carlo (SMC) methods that traverse a sequence of intermediate (often annealed) distributions connecting a tractable prior to a complex target (Chopin, 2002; Del Moral et al., 2006; Guo et al., 2025). Annealed importance sampling (AIS) provides the correction mechanism via importance weights (Neal, 2001; Guo et al., 2026a). Early non-neural approaches learned or adapted transition kernels within this pipeline (Wu et al., 2020; Geffner & Domke, 2021; Heng et al., 2020). With the advent of learnable proposals, normalizing flows (Dinh et al., 2014; Rezende & Mohamed, 2015; Chen et al., 2018) have been used to parameterize SMC/MCMC kernels and to refine proposals via variational objectives (Arbel et al., 2021; Matthews et al., 2022). These methods typically improve sample efficiency but still rely on importance weights to ensure correct marginal sampling.

More recently, diffusion-based annealed samplers that leverage score-based generative modeling (Song et al., 2021b) have been proposed, including both non-learning-based approaches with rigorous analyses (e.g., (Huang et al., 2024; He et al., 2024)), and learning-based approaches (e.g., (Vargas et al., 2024; Akhound-Sadegh et al., 2024)). These methods use annealed dynamics derived from denoising diffusion processes and combine importance weighting with a learning objective such as score matching or LV to mitigate mismatch between simulated trajectories and the target. Sequential Control for Langevin Dynamics (SCLD) (Chen et al., 2025) improves training efficiency by off-policy optimization over annealed dynamics, although importance weighting is still required to achieve accurate marginals. A complementary direction augments diffusion models with MCMC components for sampling Boltzmann distributions (De Bortoli et al., 2024; Akhound-Sadegh et al., 2024) or adopts alternative training criteria such as action matching (Albergo & Vanden-Eijnden, 2025; Neklyudov et al., 2023). In practice, these approaches often inherit a significant computational

cost from evaluating energies to form importance weights, motivating schemes that reduce or reuse weighting while retaining correctness.

## A.2 DISCRETE DIFFUSION SAMPLERS

The study of neural samplers for continuous distributions has greatly inspired the construction of similar model structures and algorithms for discrete distributions. Holderrieth et al. (2025) proposed LEAPS that leverages the discrete version of Jarzynski equality and learns the transport via a locally equivariant network, and Ou et al. (2025b) follows this line of study and proposed discrete neural flow samplers (DNFS) with a different loss formulation. The work Zhu et al. (2025) solved the sampling problem through the SOC perspective similar to Zhang & Chen (2022), and proposed samplers based on masked and uniform discrete diffusion. There is also a line of research that leverages discrete diffusion for solving combinatorial optimization problems via sampling, e.g., Sanokowski et al. (2024; 2025). More recently, So et al. (2026); Guo et al. (2026b) generalized the adjoint sampler (Havens et al., 2025) to discrete state spaces and build samplers trained from discrete versions of adjoint matching.

## B COMPLETE ALGORITHMS FOR PROXIMAL DIFFUSION NEURAL SAMPLER

In this section, we present the complete algorithms for PDNS. We first detail the practical components; the replay buffer and the $\eta$-scheduler. We then describe several instantiations of PDNS. One class employs priority-based resampling using computed weights in (15). The other variant uniformly samples from the buffer and uses importance weights in the loss functional (see (13)). Moreover, we introduce two versions of computing weights ((8) and (10)).

**Replay Buffer** In the beginning of each stage $k$, we sample a set of trajectories $\{X^{(i)}\}_{i=1}^N \sim \mathbb{P}^{\theta_{k-1}}$ or $\mathbb{P}^{k-1}$. Then, we compute the weight

$$w_\eta^{(i)} := \left( e^{r(X_T^{(i)})} \frac{d\mathbb{P}^{\text{pref}}}{d\mathbb{P}^{\bar{\theta}_{k-1}}}(X^{(i)}) \right)^{\frac{\eta_k}{\eta_k+1}} \quad \text{or} \quad w_\lambda^{(i)} := e^{(1-\lambda_k)r(X_T^{(i)})} \frac{d\mathbb{P}^{\text{pref}}}{d\mathbb{P}^{k-1}}(X^{(i)}), \quad (17)$$

for each $i \in \{0, 1, \ldots, N\}$. To simplify the notation, we denote $w^{(i)}$ as $w_\eta^{(i)}$ or $w_\lambda^{(i)}$, depending on the local optimization problem we choose. We store a terminal state $X_T^{(i)}$ and corresponding weight $w^{(i)}$ into a buffer, i.e., $\mathcal{B} := \{(X_T^{(i)}, w^{(i)})\}_{i=1}^N$.

**Adaptive Scheduler** We provide a concrete implementation of the adaptive scheduler. Given a KL trust-region radius $\epsilon > 0$, we aim to choose the largest $\eta_k$ such that satisfies $\text{KL}(\mathbb{P}^{\theta_{k-1}} \| \mathbb{P}^{k^*}) \leq \epsilon$. With the samples in the replay buffer $\mathcal{B} := \{(X_T^{(i)}, w^{(i)})\}_{i=1}^N$, the KL divergence can be estimated via empirical measures, $\mathbb{P}^{\theta_{k-1}} \approx \frac{1}{N}\sum_{i=1}^N \delta_{X^{(i)}}, \mathbb{P}^{k^*} \approx \frac{1}{N}\sum_{i=1}^N w_\eta^{(i)}\delta_{X^{(i)}}$, as follows:

$$\text{KL}(\mathbb{P}^{\theta_{k-1}} \| \mathbb{P}^{k^*}) = -\int \log \frac{d\mathbb{P}^{k^*}}{d\mathbb{P}^{\theta_{k-1}}}(x)d\mathbb{P}^{\theta_{k-1}}(x) \approx -\frac{1}{N}\sum_{i=1}^N \log(Nw^{(i)}(\eta)), \quad (18)$$

$w^{(i)}(\eta)$ is the weight for $X^{(i)}$ computed through (17), and here we emphasize its dependence on $\eta$. We then select $\eta_k$ by solving the one-dimensional least-squares to the trust-region constraint $\epsilon$:

$$\eta_k := \underset{\eta}{\arg\min} \left( \epsilon + \frac{1}{N}\sum_{i=1}^N \log(N\hat{w}^{(i)}(\eta)) \right)^2. \quad (19)$$

In practice, we optimize this objective with L-BFGS optimizer (Liu & Nocedal, 1989). Note that the computation is negligible compared to model updates.

**Resampling-based & Weight-based Algorithms** At stage $k$, let the buffer be

$$\mathcal{B} := \{(X_T^{(i)}, w^{(i)})\}_{i=1}^N.$$

Define normalized weights $\bar{w}^{(i)} := w^{(i)} / \sum_{j=1}^N w^{(j)}$. Then the empirical approximation of $\mathbb{P}_T^{\theta_k^*}$ (or $\mathbb{P}_T^{k^*}$) is

$$\widehat{\mathbb{P}}_T = \sum_{i=1}^N \bar{w}^{(i)} \delta_{X_T^{(i)}}.$$

We can instantiate the CE loss in two equivalent ways:

**Weight-based:**

$$\mathcal{L}_{\mathrm{w}}(\theta) \; = \; -\sum_{i=1}^{N} \bar{w}^{(i)} \log \mathbb{P}^{\theta}\big(X^{(i)}\big). \tag{20}$$

**Resampling-based:** draw indices $I_1, \ldots, I_N \sim \mathrm{Categorical}(\bar{w}^{(1)}, \ldots, \bar{w}^{(N)})$ and set $\hat{X}^{(i)} := X^{(I_i)}$. Then use the unweighted loss

$$\mathcal{L}_{\mathrm{rs}}(\theta) \; = \; -\frac{1}{N}\sum_{i=1}^{N} \log \mathbb{P}^{\theta}\big(\hat{X}^{(i)}\big). \tag{21}$$

We refer to these as the *weight-based* and *resampling-based* algorithms, respectively. For both continuous and discrete diffusion models, we can replace the log-likelihood in above two equations by matching losses. Let the score matching (Song et al., 2021b) or bridge matching (Shi et al., 2023) loss be denote as $\mathcal{L} : \mathcal{X} \to \mathbb{R}$. Then, the corresponding WDCE variants of weight- and resample-based methods are provided in Algorithms 2 and 3, respectively.

---

**Algorithm 2** Proximal Diffusion Neural Sampler (PDNS) v1

---

**Require:** Model parameters $\theta$, target distribution $\pi \propto e^{-\beta E}$, reference path measure $\mathbb{P}^{\mathrm{ref}}$, initial path measure $\mathbb{P}^{\theta_0}$, Buffer $\mathcal{B}$ with buffer size $N$, batch size $B$, number of inner iterations $L$.
1: **for** $k$ **in** $1, 2, ..., L$ **do**
2:      Sample a buffer $\mathcal{B} := \{(X_T^{(i)}, w^{(i)})\}_{i=1}^{N}$ by using an predefined/adaptive scheduler.
3:      **for** step **in** $1, 2, ..., L$ **do**
4:          Randomly select $B$ number of indices $I \subset \{1, \ldots, N\}$.
5:          Compute the loss $\sum_{i \in I} w^{(i)} \mathcal{L}(X_T^{(i)})$ and update $\theta$.
6:      **end for**
7:      Set $\mathbb{P}^{\theta_k} \leftarrow \mathbb{P}^{\theta}$.
8: **end for**

---

---

**Algorithm 3** Proximal Diffusion Neural Sampler (PDNS) v2

---

**Require:** Model parameters $\theta$, target distribution $\pi \propto e^{-\beta E}$, reference path measure $\mathbb{P}^{\mathrm{ref}}$, initial path measure $\mathbb{P}^{\theta_0}$, Buffer $\mathcal{B}$ with buffer size $N$, batch size $B$, number of inner iterations $L$.
1: **for** $k$ **in** $1, 2, ..., L$ **do**
2:      Sample a buffer $\mathcal{B} := \{(X_T^{(i)}, w^{(i)})\}_{i=1}^{N}$ by using an predefined/adaptive scheduler.
3:      Resample $N$ numbers of tuple in the buffer $\mathcal{B}$ proportional to weights $\{w^{(i)}\}$, i.e.,

$$J = \mathrm{Cat}(\{w^{(i)}\}), \quad \hat{\mathcal{B}} := \{X_T^{(i)}\}_{i \in J}.$$

4:      **for** step **in** $1, 2, ...$ **do**
5:          Randomly select $B$ number of samples $\{\hat{X}_T^{(i)}\}_{i=1}^{B}$ in $\hat{\mathcal{B}}$.
6:          Compute the loss $\frac{1}{N}\sum_{i=1}^{B} \mathcal{L}(\hat{X}_T^{(i)})$ and update $\theta$.
7:      **end for**
8:      Set $\mathbb{P}^{\theta_k} \leftarrow \mathbb{P}^{\theta}$.
9: **end for**

---

Table 7: Comparison between weight-based (Alg. 2) and resampling-based (Alg. 3) method.

| Target | DW-4 | | LJ-13 | | Ising, $L = 8, J = 1, \beta_{\mathrm{low}} = 0.6$ | | |
|---|---|---|---|---|---|---|---|
| Metric | $\mathcal{W}_2$ ($\downarrow$) | $E(\cdot)\,\mathcal{W}_2$ ($\downarrow$) | $\mathcal{W}_2$ ($\downarrow$) | $E(\cdot)\,\mathcal{W}_2$ ($\downarrow$) | ESS ($\uparrow$) | $E(\cdot)\,\mathcal{W}_2$ ($\downarrow$) | Adaptive Training Steps ($\downarrow$) |
| weight | 0.50 | 0.32 | 1.52 | 5.96 | 0.984 | 1.052 | 3080 |
| Resampling | 0.51 | 0.21 | 1.57 | 1.01 | 0.972 | 0.844 | 4339 |

**Comparison between Weight- and Resample-based Algorithm** We compare two instantiation of our method; weight- and resample-based methods. As shown in Tab. 7, reweighting and resampling

achieve comparable performance in terms of the 2-Wasserstein distance $\mathcal{W}_2$. However, resampling consistently yields better performance on the energy-weighted Wasserstein metric $E(\cdot)\,\mathcal{W}_2$ in all tasks, across all tasks, albeit at the cost of slower convergence when using the adaptive scheduler.

## C THEORY OF PROXIMAL CROSS ENTROPY AND THE PROOFS

### C.1 PROOFS

**Proof of (3).**

*Proof.* As $\mathbb{P}_0^\theta = \mathbb{P}_0^{\text{ref}} = \mu$, by the chain rule of KL divergence, we have

$$
\begin{aligned}
&- \mathbb{E}_{\mathbb{P}^\theta(X)}\, r(X_T) + \text{KL}(\mathbb{P}^\theta \| \mathbb{P}^{\text{ref}}) \\
&= - \mathbb{E}_{\mathbb{P}^\theta(X)}\, r(X_T) + \mathbb{E}_{\mu(X_0)}\, \text{KL}(\mathbb{P}_{(0,T]|0}^\theta(\cdot|X_0) \| \mathbb{P}_{(0,T]|0}^{\text{ref}}(\cdot|X_0)) \\
&= \mathbb{E}_{\mu(X_0)}\, \mathbb{E}_{\mathbb{P}_{(0,T]|0}^\theta(X_{(0,T]}|X_0)} \left[ -r(X_T) + \log \frac{\mathbb{P}_{(0,T]|0}^\theta(X_{(0,T]}|X_0)}{\mathbb{P}_{(0,T]|0}^{\text{ref}}(X_{(0,T]}|X_0)} \right] \\
&= \mathbb{E}_{\mu(X_0)}\, \mathbb{E}_{\mathbb{P}_{(0,T]|0}^\theta(X_{(0,T]}|X_0)} \log \frac{\mathbb{P}_{(0,T]|0}^\theta(X_{(0,T]}|X_0)}{\mathbb{P}_{(0,T]|0}^{\text{ref}}(X_{(0,T]}|X_0)\mathrm{e}^{r(X_T)}}.
\end{aligned}
$$

Therefore, we have

$$
\mathbb{P}_{(0,T]|0}^*(X_{(0,T]}|X_0) \propto \mathbb{P}_{(0,T]|0}^{\text{ref}}(X_{(0,T]}|X_0)\mathrm{e}^{r(X_T)}.
$$

Let the normalizing constant on the right-hand-side be

$$
\mathrm{e}^{V_0(X_0)} := \mathbb{E}_{\mathbb{P}_{(0,T]|0}^{\text{ref}}(X_{(0,T]}|X_0)}\,\mathrm{e}^{r(X_T)} = \mathbb{E}_{\mathbb{P}_{T|0}^{\text{ref}}(X_T|X_0)}\,\mathrm{e}^{r(X_T)},
$$

which implies

$$
\mathbb{P}_{(0,T]|0}^*(X_{(0,T]}|X_0) = \mathbb{P}_{(0,T]|0}^{\text{ref}}(X_{(0,T]}|X_0)\mathrm{e}^{r(X_T)-V_0(X_0)} \implies \mathbb{P}^*(X) = \mathbb{P}^{\text{ref}}(X)\mathrm{e}^{r(X_T)-V_0(X_0)}.
$$

Thus, by integrating out $X_{(0,T)}$, we have

$$
\mathbb{P}_{0,T}^*(X_0, X_T) = \mathbb{P}_{0,T}^{\text{ref}}(X_0, X_T)\mathrm{e}^{r(X_T)-V_0(X_0)}. \tag{22}
$$

Due to the memoryless condition,

$$
\mathrm{e}^{V_0(X_0)} = \mathbb{E}_{\mathbb{P}_T^{\text{ref}}(X_T)}\,\mathrm{e}^{r(X_T)} = \mathbb{E}_\nu\, \frac{\mathrm{e}^{-\beta E}}{\nu} = Z,
$$

which is the normalizing constant of $\pi \propto \mathrm{e}^{-\beta E}$. Therefore, we have

$$
\mathbb{P}_{0,T}^*(X_0, X_T) = \mu(X_0)\nu(X_T)\frac{\mathrm{e}^{-\beta E(X_T)}}{\nu(X_T)}\frac{1}{Z} = \mu(X_0)\pi(X_T),
$$

which completes the proof. $\qquad\square$

**Remark.** If the memoryless assumption of $\mathbb{P}^{\text{ref}}$ does not hold, then from (22), we have

$$
\begin{aligned}
\mathbb{P}_T^*(X_T) &= \int \mathbb{P}_{0,T}^{\text{ref}}(X_0, X_T)\mathrm{e}^{r(X_T)-V_0(X_0)}\mathrm{d}X_0 \\
&= \int \underbrace{\mathbb{P}_{0|T}^{\text{ref}}(X_0|X_T)}_{\neq \mathbb{P}_0^{\text{ref}}(X_0)}\,\mathrm{e}^{-V_0(X_0)}\mathrm{d}X_0 \cdot \underbrace{\mathbb{P}_T^{\text{ref}}(X_T)\mathrm{e}^{r(X_T)}}_{=Z\pi(X_T)},
\end{aligned}
$$

which is intractable and typically not equal to $\pi$, i.e., the optimization problem (3) would not lead to a path measure that ends at our target distribution $\pi$. We refer readers to Domingo-Enrich et al. (2025) for further discussion on the memoryless assumption.

**Proof of Prop. 3.1.**

*Proof.* **1.** Similar to the proof of (3), we have

$$- \mathbb{E}_{\mathbb{P}^\theta(X)} \, r(X_T) + \mathrm{KL}(\mathbb{P}^\theta \| \mathbb{P}^{\mathrm{ref}}) + \frac{1}{\eta_k} \mathrm{KL}(\mathbb{P}^\theta \| \mathbb{P}^{\theta_{k-1}})$$

$$= \mathbb{E}_{\mathbb{P}^\theta(X)} \left[ -r(X_T) + \log \frac{\mathbb{P}^\theta(X)}{\mathbb{P}^{\mathrm{ref}}(X)} + \log \frac{(\mathbb{P}^\theta(X))^{1/\eta_k}}{(\mathbb{P}^{\theta_{k-1}}(X))^{1/\eta_k}} \right]$$

$$= \mathbb{E}_{\mathbb{P}^\theta(X)} \log \frac{(\mathbb{P}^\theta(X))^{(1+\eta_k)/\eta_k}}{\mathbb{P}^{\mathrm{ref}}(X) e^{r(X_T)} (\mathbb{P}^{\theta_{k-1}}(X))^{1/\eta_k}}$$

$$= \frac{1+\eta_k}{\eta_k} \mathbb{E}_{\mathbb{P}^\theta} \log \frac{\mathbb{P}^\theta}{(\mathbb{P}^*)^{\eta_k/(1+\eta_k)} (\mathbb{P}^{\theta_{k-1}})^{1/(1+\eta_k)}}$$

$$= \frac{1+\eta_k}{\eta_k} \mathrm{KL}(\mathbb{P}^\theta \| \mathbb{P}^{\theta_k^*}) + \mathrm{const.}$$

**2.** The first claim is easy to prove by induction. To obtain the second claim, it suffices to note the relation $\mathbb{P}^*(X) \propto \mathbb{P}^{\mathrm{ref}}(X) e^{r(X_T)}$ where $r = \log \frac{\pi}{\nu} + \mathrm{const.}$ $\square$

## C.2 Abstract form to Continuous Diffusion Models

### C.2.1 SOC Formulation for Sampling Problem

**Sampling Problem** Suppose $\pi(x) \propto \exp(-\beta E(x))$ for $x \in \mathcal{X} \subseteq \mathbb{R}^d$. Suppose that the reference dynamics is given as follows:

$$\mathrm{d}X_t = b_t(X_t)\mathrm{d}t + \sigma_t \mathrm{d}W_t, \quad X_0 \sim \mu. \tag{23}$$

Let $\mathbb{P}^{\mathrm{ref}}$ be the path measure induced by (23). Let $r(x) = -\beta E(x) - \log \nu(x)$, where $\nu := \mathbb{P}_T^{\mathrm{ref}}$. Our goal is to obtain a sample $x$ from the target distribution $\pi$, i.e.,

$$x \sim \pi = \frac{\mathrm{d}\pi}{\mathrm{d}\nu} \nu \propto e^{r(\cdot)} \mathbb{P}_T^{\mathrm{base}}(\cdot). \tag{24}$$

**SOC-based Diffusion Samplers** Assume that the reference dynamics are *memoryless*, meaning that the random variables $X_0$ and $X_1$ generated from the reference process (23) are independent, i.e.

$$\mathbb{P}_{0,T}^{\mathrm{ref}}(X_0, X_T) = \mathbb{P}_0^{\mathrm{ref}}(X_0)\mathbb{P}_T^{\mathrm{ref}}(X_T). \tag{25}$$

Given this setting, we now consider the following Stochastic Optimal Control (SOC) problem:

$$\min_u \mathbb{E}_{X \sim p^u} \left[ \int_0^T \frac{1}{2} \|u_t(X_t)\|^2 \mathrm{d}t - r(X_T) \right]$$

$$\text{s.t. } \mathrm{d}X_t = (b_t(X_t) + \sigma_t u_t(X_t)) \, \mathrm{d}t + \sigma_t \mathrm{d}W_t, \; X_0 \sim \mu. \tag{26}$$

Let $\mathbb{P}^u$ be the path measure induced by the controlled dynamics and let $u^*$ be the optimal control. Then, the optimal path measure $\mathbb{P}^* := \mathbb{P}^{u^*}$ can be analyzed as follows (Domingo-Enrich et al., 2025; Liu et al., 2025):

$$\mathbb{P}_T^*(X_T) = \int \mathbb{P}_{0,T}^{\mathrm{ref}}(X_0, X_T) e^{r(X_T) + v_0(X_0)} \mathrm{d}X_0, \quad v_0(x) = -\log \int \mathbb{P}_{T|0}^{\mathrm{ref}}(y|x) e^{r(y)} \mathrm{d}y. \tag{27}$$

Due to the memoryless condition, $v_0 \equiv \mathrm{const}$ there by

$$\mathbb{P}_T^*(X_T) = \int \mu(X_0)\nu(X_T) e^{r(X_T)} \mathrm{d}X_0 = \nu(X_T)e^{r(X_T)} \propto \pi(X_T). \tag{28}$$

Thus, The terminal distribution matches our target distribution, i.e., $\mathbb{P}_1^* = \pi$. Note that we call the objective in (26) as an *relative entropy* (RE) loss.

**Cross-Entropy (CE)** Cross-entropy counterpart of (26) can be defined as follows:

$$\min_u \mathrm{KL}(\mathbb{P}^* \| \mathbb{P}^u). \tag{29}$$

Now, we decompose this cross-entropy objective into the tractable form:

$$\text{KL}(\mathbb{P}^*\|\mathbb{P}^u) = \mathbb{E}_{X\sim\mathbb{P}^*}\left[\log\frac{d\mathbb{P}^*}{d\mathbb{P}^u}(X)\right] = -\mathbb{E}_{X\sim\mathbb{P}^*}\left[\log\mathbb{P}^u(X)\right] + C \tag{30}$$

$$= -\mathbb{E}_{X\sim\mathbb{P}^v}\left[\frac{d\mathbb{P}^*(X)}{d\mathbb{P}^v(X)}\log\mathbb{P}^u(X)\right] + C, \tag{31}$$

where $v$ is arbitrarily given control. Note that we need an auxiliary control $v$ to sample a trajectory since we do not know the optimal path measure $\mathbb{P}^*$. Instead, we need to compute Radon-Nikodym (RN) derivative:

$$\frac{d\mathbb{P}^*}{d\mathbb{P}^v}(X) = \frac{d\mathbb{P}^*}{d\mathbb{P}^{\text{ref}}}(X)\frac{d\mathbb{P}^{\text{ref}}}{d\mathbb{P}^v}(X) \overset{memoryless}{\propto} \exp\left(\int_0^T -\frac{1}{2}\|v_t(X_t)\|^2 dt - v_t(X_t)\cdot dW_t + r(X_T)\right). \tag{32}$$

### C.2.2 Weighted Denoising Cross-Entropy (WDCE)

**Reciprocal Property** The *reciprocal property* (Léonard, 2012; Léonard et al., 2014; Chen et al., 2021; Shi et al., 2024) implies that

$$\mathbb{P}^*(X) = \mathbb{P}^*_T(X_T)\mathbb{P}^{\text{ref}}_{\cdot|T}(X|X_T) = \mathbb{P}^*_{0,T}(X_0, X_T)\mathbb{P}^{\text{ref}}_{\cdot|0,T}(X|X_0, X_T). \tag{33}$$

When the reference dynamics is *memoryless*,

$$\mathbb{P}^*_{0,T}(X_0, X_T) \propto \mu(X_0)\nu(X_T)e^{r(X_T)} \propto \mu(X_0)\pi(X_T), \tag{34}$$

which implies that $\mathbb{P}^*$ is memoryless, i.e., the joint marginal distribution $\mathbb{P}^*_{0,T}(X_0, X_1) = \mu(X_0)\pi(X_T)$ is independent. Combining (33) and (34), we obtain

$$\mathbb{P}^*(X) = \mu(X_0)\mathbb{P}^*_T(X_T)\mathbb{P}^{\text{ref}}_{\cdot|0,T}(X|X_0, X_T). \tag{35}$$

**WDCE** The following SDE induces the optimal path measure $\mathbb{P}^*$

$$dX_t = \left(b_t(X_t) + \sigma_t^2\,\mathbb{E}_{\mathbb{P}^*}\left[\nabla_{x_t}\log\mathbb{P}^{\text{ref}}_{T|t}(X_T|X_t)\right]\right)dt + \sigma_t dW_t, \quad X_0 \sim \mu. \tag{36}$$

using Doob's $h$-transform theory (Rogers & Williams, 2000). Note that this is a generalization of score matching (Anderson, 1982; Song et al., 2021b). Then, our CE objective (29) can be extended to denoising CE as follows:

$$\underset{u}{\arg\min}\,\text{KL}(\mathbb{P}^*\|\mathbb{P}^u) = \underset{u}{\arg\min}\,\mathbb{E}_{X\sim\mathbb{P}^*}\left[\int_0^T\frac{1}{2}\|u_t(X_t) - \sigma_t\nabla\log\mathbb{P}^*_{T|t}(X_T|X_t)\|^2 dt\right]. \tag{37}$$

Finally, we obtain the **weighted denoising cross entropy** optimization problem:

$$u^* = \underset{u}{\arg\min}\,\mathbb{E}_{X\sim\mathbb{P}^*}\left[\int_0^T\frac{1}{2}\|u_t(X_t) - \sigma_t\nabla\log\mathbb{P}^*_{T|t}(X_T|X_t)\|^2 dt\right] \tag{38}$$

$$= \underset{u}{\arg\min}\,\mathbb{E}_{X_T\sim\mathbb{P}^*_T}\,\mathbb{E}_{X_t|X_T\sim\mathbb{P}^{\text{ref}}_{t|T}}\left[\int_0^T\frac{1}{2}\|u_t(X_t) - \sigma_t\nabla\log\mathbb{P}^*_{T|t}(X_T|X_t)\|^2 dt\right] \tag{39}$$

$$= \underset{u}{\arg\min}\,\mathbb{E}_{X_T\sim\mathbb{P}^v}\,\mathbb{E}_{X_t|X_T\sim\mathbb{P}^{\text{ref}}_{t|T}}\left[\frac{d\mathbb{P}^*}{d\mathbb{P}^v}(X)\int_0^T\frac{1}{2}\|u_t(X_t) - \sigma_t\nabla\log\mathbb{P}^*_{T|t}(X_T|X_t)\|^2 dt\right]. \tag{40}$$

Therefore, combining we can run WDCE by the following iterative process:

1. Sample $N$ trajectories $\{X^{(i)}\}_{i=1}^N \sim \mathbb{P}^v$.
2. Compute weights $\{w^{(i)}\}_{i=1}^N$ by (32) for each corresponding trajectory $\{X^{(i)}\}_{i=1}^N$.
3. Resample $\{X_1^{(i)}\}_{i=1}^N$ by following categorical distribution:

$$\{\hat{X}_1^{(i)}\}_{i=1}^N \sim \text{Cat}\left(\{\hat{w}^{(i)}\}_{i=1}^N, \{X_1^{(i)}\}_{i=1}^N\right), \quad \text{where } \hat{w}^{(i)} = \frac{w^{(i)}}{\sum_{i=1}^N w^{(i)}}. \tag{41}$$

4. Update the control $u_\theta := u$ through a *score matching* loss with a resampled data.

**Remark.** Particle Denosing Diffusion Sampler (PDDS) (Phillips et al., 2024) is one of the sampling method which leverages this Importance weighted CE method.

### C.2.3 Theories on Proximal Diffusion Neural Samplers

In this section, we introduce the theory for PDNS written in SDE formulations.

**Proximal Relative Entropy**      Suppose we have the current control $u^{k-1}$ and the corresponding path measure $\mathbb{P}^{k-1}$. Then, consider the following iterative method:

$$
\begin{aligned}
u^k &= \operatorname*{argmin}_{u} \left[ \mathbb{E}_{X \sim \mathbb{P}^u} \left[ -r(X_T) \right] + \mathrm{KL}(\mathbb{P}^u | \mathbb{P}^{\mathrm{ref}}) + \frac{1}{\eta_k} \mathrm{KL}(\mathbb{P}^u | \mathbb{P}^{k-1}) \right] \\
&= \operatorname*{argmin}_{u} \mathbb{E}_{X \sim \mathbb{P}^u} \left[ \int_0^T \left[ \frac{1}{2} \|u_t(X_t)\|^2 + \frac{1}{2\eta_k} \|u_t(X_t) - u^{k-1}(X_t)\|^2 \right] \mathrm{d}t - r(X_T) \right].
\end{aligned}
\tag{42}
$$

Note that we denote (42) as a *proximal RE* (Zhang et al., 2026).

**Proximal Cross Entropy**      Solving the Proximal RE problem in (42) is computationally expensive, as it requires simulating trajectories under the current (online) policy and solving a numerical optimization problem, often via the adjoint system (Yong & Zhou, 1999; Nüsken & Richter, 2021; Domingo-Enrich et al., 2025; Havens et al., 2025). To mitigate this cost, we reformulate the Proximal RE objective as a cross-entropy problem by (11) and (12):

$$
\mathrm{KL}(\mathbb{P}^{\theta_k^*} \| \mathbb{P}^\theta) \propto - \mathbb{E}_{X \sim \mathbb{P}^{\bar{\theta}_{k-1}}} \left( \mathrm{e}^{r(X_T)} \frac{\mathrm{d}\mathbb{P}^{\mathrm{ref}}}{\mathrm{d}\mathbb{P}^{\bar{\theta}_{k-1}}}(X) \right)^{\frac{\eta_k}{\eta_k+1}} \log \mathbb{P}^\theta(X) + C,
\tag{43}
$$

$$
\mathrm{KL}(\mathbb{P}^k \| \mathbb{P}^\theta) \propto - \mathbb{E}_{X \sim \mathbb{P}^{k-1}} \left( \mathrm{e}^{(1-\lambda_k)r(X_T)} \frac{\mathrm{d}\mathbb{P}^{\mathrm{ref}}}{\mathrm{d}\mathbb{P}^{k-1}}(X) \right) \log \mathbb{P}^\theta(X) + C,
\tag{44}
$$

Now, let $u_t^{\bar{\theta}_{k-1}}(x)$ (resp. $u_t^{k-1}(x)$) be the control that induces the path measure $\mathbb{P}^{\theta_{k-1}}$ (resp. $\mathbb{P}^{k-1}$). Using the Girsanov theorem (Øksendal, 2003; Särkkä & Solin, 2019), i.e.,

$$
\frac{\mathrm{d}\mathbb{P}^{\mathrm{ref}}}{\mathrm{d}\mathbb{P}^{\bar{\theta}_{k-1}}} = \exp \left( - \int_0^T \frac{1}{2} \|u_t^{\bar{\theta}_{k-1}}(X_t)\|^2 \mathrm{d}t + u_t^{\bar{\theta}_{k-1}}(X_t) \cdot \mathrm{d}W_t \right),
\tag{45}
$$

Precisely, the weights in (43) and (44) can be computed as follows:

$$
\begin{aligned}
& \left( \mathrm{e}^{r(X_T)} \frac{\mathrm{d}\mathbb{P}^{\mathrm{ref}}}{\mathrm{d}\mathbb{P}^{\bar{\theta}_{k-1}}}(X) \right)^{\frac{\eta_k}{\eta_k+1}} \\
& \qquad = \exp \left( -\frac{\eta_k}{1+\eta_k} \left[ \int_0^T \frac{1}{2} \|u_t^{k-1}(X_t)\|^2 \mathrm{d}t + u_t^{k-1}(X_t) \cdot \mathrm{d}W_t - r(X_1) \right] \right),
\end{aligned}
\tag{46}
$$

and

$$
\begin{aligned}
& \mathrm{e}^{(1-\lambda_k)r(X_T)} \frac{\mathrm{d}\mathbb{P}^{\mathrm{ref}}}{\mathrm{d}\mathbb{P}^{k-1}}(X) \\
& \qquad = \exp \left( - \left[ \int_0^T \frac{1}{2} \|u_t^{k-1}(X_t)\|^2 \mathrm{d}t + u_t^{k-1}(X_t) \cdot \mathrm{d}W_t - (1-\lambda_k)r(X_1) \right] \right).
\end{aligned}
\tag{47}
$$

**Proximal WDCE**      Similar to the derivation of (38), we can easily derive

$$
u^{\theta_k^*} = \mathbb{E}_{X \sim \mathbb{P}^{\bar{\theta}_k^*}} \left[ \int_0^T \frac{1}{2} \|u_t^\theta(X_t) - \sigma_t \nabla \log \mathbb{P}^{\mathrm{ref}}_{T|t}(X_T|X_t)\|^2 \mathrm{d}t \right].
\tag{48}
$$

By leveraging *reciprocal property*, i.e., $\mathbb{P}^{\theta_k^*}(X) = \mu(X_0)\mathbb{P}_T^{\theta_k^*}(X_T)\mathbb{P}^{\mathrm{ref}}_{\cdot|0,T}(X|X_0,X_T)$, we can further analyze the PCE objectives:

$$
u^{\theta_k^*} = \operatorname*{argmin}_{u} \mathbb{E}_{X_0 \sim \mu, X_T \sim \mathbb{P}^{\theta_k^*}} \mathbb{E}_{X \sim \mathbb{P}^{\mathrm{ref}}_{\cdot|0,T}(\cdot|X_0,X_T)} \left[ \int_0^T \frac{1}{2} \|u_t^\theta(X_t) - \sigma_t \nabla \log \mathbb{P}^{\mathrm{ref}}_t(X_t)\|^2 \mathrm{d}t \right].
\tag{49}
$$

By applying denosing score (bridge) matching objective (Song et al., 2021a; Shi et al., 2024), we obtain our proximal WDCE objective:

$$\mathcal{F}(\mathbb{P}^\theta; \mathbb{P}^{\theta_k^*}) = \underset{\substack{t \sim \mathrm{Unif}(0,T) \\ X_0 \sim \mu \\ X_T \sim \mathbb{P}_T^{\theta_k^*}}}{\mathbb{E}} \underset{X_t \sim \mathbb{P}_{t|0,T}^{\mathrm{ref}}(\cdot|X_0,X_T)}{\mathbb{E}} \left[ \frac{1}{2} \| u_t^\theta(X_t) - \sigma_t \nabla \log \mathbb{P}_{T|t}^{\mathrm{ref}}(X_T|X_t) \|^2 \right] \tag{50}$$

$$= \underset{\substack{t \sim \mathrm{Unif}(0,T) \\ X_0 \sim \mu \\ X_T \sim \mathbb{P}_T^{\theta_{k-1}}}}{\mathbb{E}} \underset{X_t \sim \mathbb{P}_{t|0,T}^{\mathrm{ref}}(\cdot|X_0,X_T)}{\mathbb{E}} \left[ \frac{\mathrm{d}\mathbb{P}^{\theta_k^*}}{\mathrm{d}\mathbb{P}^{\bar\theta_{k-1}}}(X) \frac{1}{2} \| u_t^\theta(X_t) - \sigma_t \nabla \log \mathbb{P}_{T|t}^{\mathrm{ref}}(X_T|X_t) \|^2 \right]. \tag{51}$$

### C.2.4 PROXIMAL WDCE IN PRACTICE

**Resampling-based Algorithm** We instantiate Proximal WDCE via (50). To obtain(50). First, to obtain a sample $X_T \sim \mathbb{P}_T^{\theta_k^*}$ required in (13), we apply importance resampling:

1. Draw $N$ trajectories $\{X^{(i)}\}_{i=1}^N$ from $\mathbb{P}^{\theta_{k-1}}$;
2. Compute weights

$$w_{\eta_k}^{\theta_k^*}(X) = \left( \mathrm{e}^{r(X_T)} \frac{\mathrm{d}\mathbb{P}^{\mathrm{ref}}}{\mathrm{d}\mathbb{P}^{\bar\theta_{k-1}}}(X) \right)^{\frac{\eta_k}{\eta_k+1}} = (46); \tag{52}$$

3. resample $\{X_T^{(i)}\}_{i=1}^N$ by following categorical distribution:

$$\{\tilde X_T\}_{i=1}^N \sim \mathrm{Cat} \left( \left\{ \frac{w_{\eta_k}^{\theta_k}(X^{(i)})}{\sum_{i=1}^N w_{\eta_k}^{\theta_k}(X^{(i)})} \right\}_{i=1}^N, \{X_T^{(i)}\}_{i=1}^N \right). \tag{53}$$

**Weighting-based Algorithm** Alternatively, using (51), we draw $X_T \sim \mathbb{P}_T^{\theta_{k-1}}$ and and incorporate $w_{\eta_k}^{\theta_k^*}$ directly in the objective, i.e., optimize a weighted loss where each sample contributes proportionally to its importance weight rather than being resampled.

### C.3 ABSTRACT FORM TO DISCRETE DIFFUSION MODELS

### C.3.1 SOC FORMULATION FOR SAMPLING PROBLEM

**Sampling Problem** We suppose the target distribution is supported on $\mathcal{X}_0 = \{1, 2, ..., N\}^d$ and let the mask-augmented state space be $\mathcal{X} = \{1, 2, ..., N, \mathbf{M}\}^d$. Let the target distribution be $\pi(x) \propto \mathrm{e}^{-\beta E}$ supported on $\mathcal{X}_0$.

The reference path measure $\mathbb{P}^{\mathrm{ref}}$ has a generator $Q^{\mathrm{ref}}$ defined by

$$Q_t^{\mathrm{ref}}(x,y) = \begin{cases} \frac{\gamma(t)}{N} & \text{if } x^i = \mathbf{M}, y = x^{i\leftarrow n}, n \neq \mathbf{M}; \\ -\gamma(t)|\{i : x^i = \mathbf{M}\} & \text{if } x = y; \\ 0 & \text{if otherwise.} \end{cases}$$

$\gamma : [0, T] \to \mathbb{R}_+$ can be any function satisfying $\int_0^T \gamma(t)\mathrm{d}t = \infty$. By construction, $\mathbb{P}_0^{\mathrm{ref}} = \mu = p_{\mathrm{mask}}$ is the delta distribution on the fully masked sequence, and $\mathbb{P}_T^{\mathrm{ref}} = \nu = p_{\mathrm{unif}}$ is the uniform distribution on $\mathcal{X}_0$. Let $r = -\beta E$.

Consider the following SOC problem:

$$\min_\theta \mathbb{E}_{X \sim \mathbb{P}^\theta} \left[ \int_0^T \sum_{y \neq X_t} \left( Q_t^\theta \log \frac{Q_t^\theta}{Q_t^{\mathrm{ref}}} - Q_t^\theta + Q_t^{\mathrm{ref}} \right)(X_t, y)\mathrm{d}t - r(X_T) \right],$$

$$\text{s.t. } X = (X_t)_{t\in[0,T]} \text{ is a CTMC on } \mathcal{X} \text{ with generator } Q^\theta, \ X_0 \sim p_{\mathrm{mask}},$$

Let the generator $Q^\theta$ be parameterized by

$$Q_t^\theta(x,y) = \begin{cases} \gamma(t)s_\theta(x)_{i,n} & \text{if } x^i = \mathbf{M}, y = x^{i\leftarrow n}, n \neq \mathbf{M}; \\ -\gamma(t)|\{i : x^i = \mathbf{M}\} & \text{if } x = y; \\ 0 & \text{if otherwise,} \end{cases}$$

where $s_\theta(x)$ is a $d \times N$ matrix whose each row is a probability vector. The optimal value of $s_\theta$ can be proved to have the following form (Zhu et al., 2025):

$$s_\theta(x)_{i,n} = \Pr_{X \sim \pi} \left( X^i = n | X^{\mathrm{UM}} = x^{\mathrm{UM}} \right), \text{ if } x^i = \mathbf{M}.$$

### C.3.2 Weighted Denoising Cross-entropy

The WDCE loss is as follows:

$$\mathrm{KL}(\mathbb{P}^{\theta_k^*} \| \mathbb{P}^\theta) = \mathbb{E}_{X \sim \mathbb{P}^{\theta_k^*}} \mathbb{E}_{\lambda \sim \mathrm{Unif}(0,1)} \left[ \frac{1}{\lambda} \mathbb{E}_{\mu_\lambda(\widetilde{x}|X_T)} \sum_{d:\widetilde{x}^d=\mathbf{M}} -\log s_\theta(\widetilde{x})_{d,X_T^d} \right],$$

$$\propto \mathbb{E}_{X \sim \mathbb{P}^{\bar\theta_{k-1}}} \left( e^{r(X_T)} \frac{d\mathbb{P}^{\mathrm{ref}}}{d\mathbb{P}^{\bar\theta_{k-1}}} \right)^{\frac{\eta_k}{\eta_k+1}} \mathbb{E}_{\lambda \sim \mathrm{Unif}(0,1)} \left[ \frac{1}{\lambda} \mathbb{E}_{\mu_\lambda(\widetilde{x}|X_T)} \sum_{d:\widetilde{x}^d=\mathbf{M}} -\log s_\theta(\widetilde{x})_{d,X_T^d} \right],$$

$$\mathrm{KL}(\mathbb{P}^k \| \mathbb{P}^\theta) = \mathbb{E}_{X \sim \mathbb{P}^k} \mathbb{E}_{\lambda \sim \mathrm{Unif}(0,1)} \left[ \frac{1}{\lambda} \mathbb{E}_{\mu_\lambda(\widetilde{x}|X_T)} \sum_{d:\widetilde{x}^d=\mathbf{M}} -\log s_\theta(\widetilde{x})_{d,X_T^d} \right],$$

$$\propto \mathbb{E}_{X \sim \mathbb{P}^{\bar\theta_{k-1}}} \left( e^{(1-\lambda_k)r(X_T)} \frac{d\mathbb{P}^{\mathrm{ref}}}{d\mathbb{P}^{\bar\theta}_{k-1}}(X) \right) \mathbb{E}_{\lambda \sim \mathrm{Unif}(0,1)} \left[ \frac{1}{\lambda} \mathbb{E}_{\mu_\lambda(\widetilde{x}|X_T)} \sum_{d:\widetilde{x}^d=\mathbf{M}} -\log s_\theta(\widetilde{x})_{d,X_T^d} \right],$$

where $\mu_\lambda(\cdot|x)$ means independently masking each entry of $x$ with probability $\lambda$. The weights can be computed by

$$\frac{d\mathbb{P}^{\mathrm{ref}}}{d\mathbb{P}^{\bar\theta_{k-1}}}(X) = \exp \left( \sum_{t:X_{t_-} \neq X_t} \log \frac{1/N}{s_{\bar\theta_{k-1}}(X_{t_-})_{i(t),X_t^{i(t)}}} \right),$$

where we assume the jump at time $t$ happens at the $i(t)$-th entry, and the total number of jumps is the sequence length $d$. The design choice of resampling-based and weighting-based algorithms can be done similarly as discussed in the continuous part above.

# D Experimental Details and Further Results for Learning Continuous Target Distributions

## D.1 Further Explanation on the Target Distributions

In this section, we describe the benchmarks used in our continuous-energy experiments. We evaluate seven targets: four synthetic distributions (MW-54, Funnel, GMM40, MoS) and three atomistic potentials (DW4, LJ13, LJ55). For the synthetic benchmarks we follow Choi et al. (2025); Chen et al. (2025). For the atomistic systems we follow Havens et al. (2025); Liu et al. (2025). For convenience, let $d$ be a data dimension. Unless noted otherwise, we use the hyperparameters in the cited sources.

**MW54** We use 5-dimension many-well potential. Precisely, we consider the energy function defined as follows:

$$E(x) = \sum_{i=1}^{d} (x_i^2 - \delta)^2, \tag{54}$$

where $x = (x_1, \ldots, x_d)$, $d = 5$, and $\delta = 4$. This leads to $2^d = 32$ wells in total.

**Funnel** We use a funnel-shaped distribution with $d = 10$. Precisely, the unnormalized density is defined as follows:

$$\mathcal{N}(x_1; 0, \sigma^2) \times \mathcal{N}((x_2, \ldots, x_{10}); 0, \exp(x_1)I), \tag{55}$$

where $x = (x_1, \ldots, x_d)$ and $\sigma = 3$.

**GMM40**  Let $m = 40$ be the number of modes for the Gaussian mixture. The target distribution $\pi$ is the equally weighted mixture

$$\pi(x) = \frac{1}{m} \sum_{i=1}^{m} \pi_i(X), \quad \pi_i = \mathcal{N}(\mu_i, I), \quad \mu_i \sim U_d(-40, 40), \tag{56}$$

where $U_d(a, b)$ is the uniform distribution over the hypercube $[a, b]^d$. This benchmark demonstrates mode exploration in high dimensions.

**MoS**  Following Blessing et al. (2024), we consider a heavy-tailed analogue with $d = 50$ and $m = 10$:

$$\pi(x) = \frac{1}{m} \sum_{i=1}^{m} \pi_i(X), \quad \pi_i(x) = t_2(x - \mu_i), \quad \mu_i \sim U_d(-10, 10), \tag{57}$$

where $t_2$ denotes the multivariate Student's $t$-distribution with two degrees of freedom and identity scale. The heavier tails make this target notably more challenging for samplers.

**DW4**  This potential was originally proposed in Köhler et al. (2020) and used in Akhound-Sadegh et al. (2024); Havens et al. (2025). It is a double-well potential with 4 particles in two dimensions, hence a total $d = 8$ state vector. We denote a state $x = [x_1; x_2; x_3; x_4] \in \mathbb{R}^d$ with $x_i \in \mathbb{R}^2$. The energy function reads

$$E(x) = \exp\left[ \frac{1}{2\tau} \sum_{i<j} \left( a(d_{ij} - d_0) + b(d_{ij} - d_0)^2 + c(d_{ij} - d_0)^4 \right) \right], \tag{58}$$

where $d_{ij} = \|x_i - x_j\|_2$ is the Euclidean distance between particles $i$ and $j$. We follow the configuration with $a = 0$, $b = -4$, $c = 0.9$, $d_0 = 1$, and temperature $\tau = 1$.

**LJ-13 and LJ-55**  The Lennard-Jones (LJ) potentials are classical intermolecular potentials commonly used in physics to model atomic interactions. Consider $n$ particles in 3-dimension space, state $x = [x_1; \ldots; x_n] \in \mathbb{R}^{3n}$ with positions $x_i \in \mathbb{R}^3$. The suffix in "LJ-13" or "LJ-55" indicates the particle count ($n = 13$ or $n = 55$). We use the following unnormalized energy:

$$E(x) = \frac{\epsilon}{2\tau} \sum_{i<j} \left[ \left( \frac{r_m}{d_{ij}} \right)^6 - \left( \frac{r_m}{d_{ij}} \right)^{12} \right] + \frac{c}{2} \sum_i \|x_i - C(x)\|^2, \tag{59}$$

where $d_{ij} = \|x_i - x_j\|_2$ is the pairwise distance and $C(x) = \frac{1}{n} \sum_{i=1}^{N} x_i$ is the center of mass. Following prior work (Havens et al., 2025; Liu et al., 2025), We use the parameter values $r_m = 1$, $\epsilon = 1$, $c = 0.5$, and $\tau = 1$. Note that the LJ-13 and LJ-55 systems correspond to $d = 39$ and $d = 165$, respectively.

### D.2  IMPLEMENTATION DETAILS

#### D.2.1  MEMORYLESS DYNAMICS IN WILD

In this section, we first specify the reference dynamics used in practice. Then, we derive closed-form expressions for (i) the reference bridge law $\mathbb{P}^{\text{ref}}_{\cdot|0,T}$, which yields the conditional distribution needed to sample $X_T$ given $(X_0, X_1)$ in (13), and (ii) the score $\nabla \log \mathbb{P}^{\text{ref}}_{T|t}$ appearing in (13). These formulas enable exact sampling from the reference bridge and analytic evaluation of the score.

**Ornstein-Uhlenbeck (OU) Process**  Throughout the continuous target benchmarks, we use OU process (Uhlenbeck & Ornstein, 1930) which is defined as follows:

$$dX_t = -\frac{\alpha_t}{2} X_t dt + \bar{\sigma}\sqrt{\alpha_t} dW_t, \quad X_0 \sim \mu := \mathcal{N}(0, \bar{\sigma}^2 I). \tag{60}$$

Note that $b_t(x) = -\frac{\alpha_t}{2} x$ and $\sigma_t = \bar{\sigma}\sqrt{\alpha_t}$. In this case, every marginal of the reference measure follows the stationary Gaussian distribution, i.e., $\mathbb{P}^{\text{ref}}_t := \mathcal{N}(0, \bar{\sigma}^2 I)$ for all $t \in [0, T]$. Thus, $\nu = \mu = \mathcal{N}(0, \bar{\sigma}^2 I)$, which allows us to use a tractable terminal reward $r(x) = -\beta E(x) - \log \nu(x)$. The conditional probability can be written as follows:

$$\mathbb{P}^{\text{ref}}_{T|0}(\cdot|x) = \mathcal{N}\left( e^{-\frac{1}{2}\int_0^T \alpha_s ds} x, \bar{\sigma}^2(1 - e^{-\frac{1}{2}\int_0^T \alpha_s ds}) I \right). \tag{61}$$

Note that when $e^{-\frac{1}{2}\int_0^T \alpha_s ds} \approx 0$, then the joint of the marginal distributions $\mathbb{P}_{0,T}^{ref}$ is independent, i.e. $\mathbb{P}^{ref}$ is memoryless. In practice, we set

$$\alpha_t := (1-t)\alpha_{min} + t\alpha_{max}, \tag{62}$$

where $(\alpha_{min}, \alpha_{max})$ is a given hyperparameter. To ensure that the reference dynamics is (almost) memoryless, we set $\alpha_{max} \gg 1$.

**Closed Form of Conditional Path Measure for OU Process** Suppose the reference dynamics is given as (60). Then the conditional probability $\mathbb{P}_{t|0,T}^{ref}$ is written as follows:

$$\mathbb{P}_{t|0,T}^{ref}(\cdot|X_0, X_T) = \mathcal{N}\left(\frac{B_t(1-C_t^2)}{1-B_T^2}X_0 + \frac{C_t(1-B_t^2)}{1-B_T^2}X_T, \ \bar{\sigma}^2\frac{(1-B_t^2)(1-C_t^2)}{1-B_T^2}I\right), \tag{63}$$

where

$$\eta_t = \int_t^T \sigma_s^2 ds, \quad B_t = \exp\left(-\frac{1}{2}\int_0^t \alpha_s ds\right), \quad C_t = \exp\left(-\frac{1}{2}\int_t^T \alpha_s ds\right).$$

**Closed Form of Conditional Score Function for OU Process** Moreover, the conditional score function $\nabla_{x_t} \ln \mathbb{P}_{T|t}^{ref}$ is written as follows:

$$\nabla_{x_t} \log \mathbb{P}_{T|t}^{ref}(X_T|X_t) = \frac{-C_t(C_t X_t - X_T)}{\bar{\sigma}^2(1-C_t^2)}. \tag{64}$$

### D.2.2 INITIALIZATION OF THE CONTROL

The typical choice for initial control would be $u^{\theta_0} \equiv 0$, i.e., $\mathbb{P}^{\theta_0} = \mathbb{P}^{ref}$. However, non-informative reference dynamics, i.e. $u_t^{\theta_0}(x) \equiv 0$, it takes a long time to achieve reasonable performance with our proximal method when the potential is in the high-dimensional sparse domain. Hence, for the continuous target benchmark, we use the informative initial control $u^{\theta_0}$ (or path measure $\mathbb{P}^{\theta_0}$) instead of starting from trivial control. To obtain a informative sample $X_T$ at the first stage, we use *annealed* dynamics (Neal, 2001; Matthews et al., 2022; Guo et al., 2025; Albergo & Vanden-Eijnden, 2025) defined as follows:

$$dX_t = -\frac{\alpha_t}{2}\nabla V_t(X_t)dt + \bar{\sigma}\sqrt{\alpha_t}dW_t, \quad X_0 \sim \mu. \tag{65}$$

We set

$$V_t(x) = (1-t)\nabla \log \mu(x) + t\nabla \log \pi(x) = (1-t)\nabla \log \mu(x) - t\beta E(x),$$

which is a typical annealed dynamics used in Albergo & Vanden-Eijnden (2025); Chen et al. (2025); Choi et al. (2025). At the first stage, we train WDCE objective with the samples $\{X_T^{(i)}\}_{i=1}^N$ obtained from the annealed dynamics. (We use uniform weights for the initial stage.) For energy $E$ with steep landscapes, we clip the energy gradient norm with predefined $(\nabla E)_{max}$ which is treated as a hyperparameter in Tab. 8.

### D.2.3 HYPERPARAMETER SETTINGS AND METRIC

**Model Backbone** For the MW54, Funnel, GMM40, and MoS experiments, we adopt the model architecture of Choi et al. (2025); Liu et al. (2025). For the DW4, LJ13, and LJ55 benchmarks, we follow the architecture in Liu et al. (2025), which employs an Equivariant Graph Neural Network (EGNN; Satorras et al. 2021).

**Hyperparameters** The hyperparameters for all experiments are provided in Tab. 8. The number of stages is iteration number of outer loop (line 1 in Alg. 1) and the number of inner epochs is the number of epochs used for inner loop (see line 3 in Alg. 1). We use Adam optimizer with no weight decay for all benchmarks.

**Evaluation Metrics** For Tab. 2, we compute MMD and entropic OT (Sinkhorn, $\varepsilon = 10^{-3}$) using the implementation of Blessing et al. (2024); baseline results are taken from Blessing et al. (2024); Chen et al. (2025); Akhound-Sadegh et al. (2024). For Table Tab. 3, DW/LJ tasks where rigid-motion and particle-permutation symmetries are relevant, we additionally include a symmetry-aware geometric $\mathcal{W}_2$ following Akhound-Sadegh et al. (2024) with the alignment heuristic of Köhler et al. (2020). All metrics use 2,000 generated and 2,000 reference samples. Note that the LJ-55 experiment requires approximately 80 hours of training on a single H100 GPU, with a GPU memory usage of around 14 GB. The total training time is comparable to ASBS (Liu et al., 2025).

Table 8: Hyperparameters for continuous target distribution benchmarks.

| Class | Hyperparameters | MW54 | Funnel | GMM40 | MoS | DW4 | LJ13 | LJ55 |
|---|---|---|---|---|---|---|---|---|
| Dimension | - | 5 | 10 | 50 | 50 | 8 | 39 | 165 |
| SDE | $\bar{\sigma}$ | 2.0 | 3.0 | 50.0 | 15.0 | 2.0 | 2.0 | 2.0 |
| | $(\alpha_{\min}, \alpha_{\max})$ | $(0.1, 10)$ | $(0.1, 10)$ | $(0.1, 5)$ | $(0.1, 10)$ | $(0.1, 10)$ | $(0.1, 10)$ | $(0.1, 10)$ |
| | $(\nabla E)_{\max}$ | 100.0 | 100.0 | 100.0 | 100.0 | 100.0 | 100.0 | 100.0 |
| | NFE | 200 | 200 | 1000 | 1000 | 1000 | 1000 | 1000 |
| Buffer & Scheduler | $\epsilon$ | 1.0 | 1.0 | 0.1 | 0.1 | 0.1 | 1.0 | 1.0 |
| | $|\mathcal{B}|$ | $10^5$ | $10^5$ | $5 \times 10^4$ | $5 \times 10^4$ | $5 \times 10^4$ | $5 \times 10^4$ | $5 \times 10^4$ |
| Training Hyps | # Stages | 20 | 20 | 10 | 10 | 30 | 50 | 20 |
| | # Inner Epochs | 50 | 200 | 5000 | 1000 | 1000 | 200 | 500 |
| | Batch Size | 500 | 500 | 2000 | 2000 | 500 | 500 | 500 |
| Optimizer & EMA | lr | $10^{-4}$ | $10^{-4}$ | $10^{-4}$ | $10^{-4}$ | $10^{-4}$ | $10^{-4}$ | $10^{-4}$ |
| | $(\beta_1, \beta_2)$ | $(0, 0.9)$ | $(0, 0.9)$ | $(0, 0.9)$ | $(0, 0.9)$ | $(0, 0.9)$ | $(0, 0.9)$ | $(0, 0.9)$ |
| | EMA decay | 0.999 | 0.999 | 0.999 | 0.999 | 0.999 | 0.999 | 0.999 |
| Model | Type | MLP | MLP | MLP | MLP | EGNN | EGNN | EGNN |
| | # Layers | 4 | 4 | 4 | 4 | 5 | 5 | 5 |
| | # Channels | 256 | 256 | 2048 | 2048 | 128 | 128 | 128 |

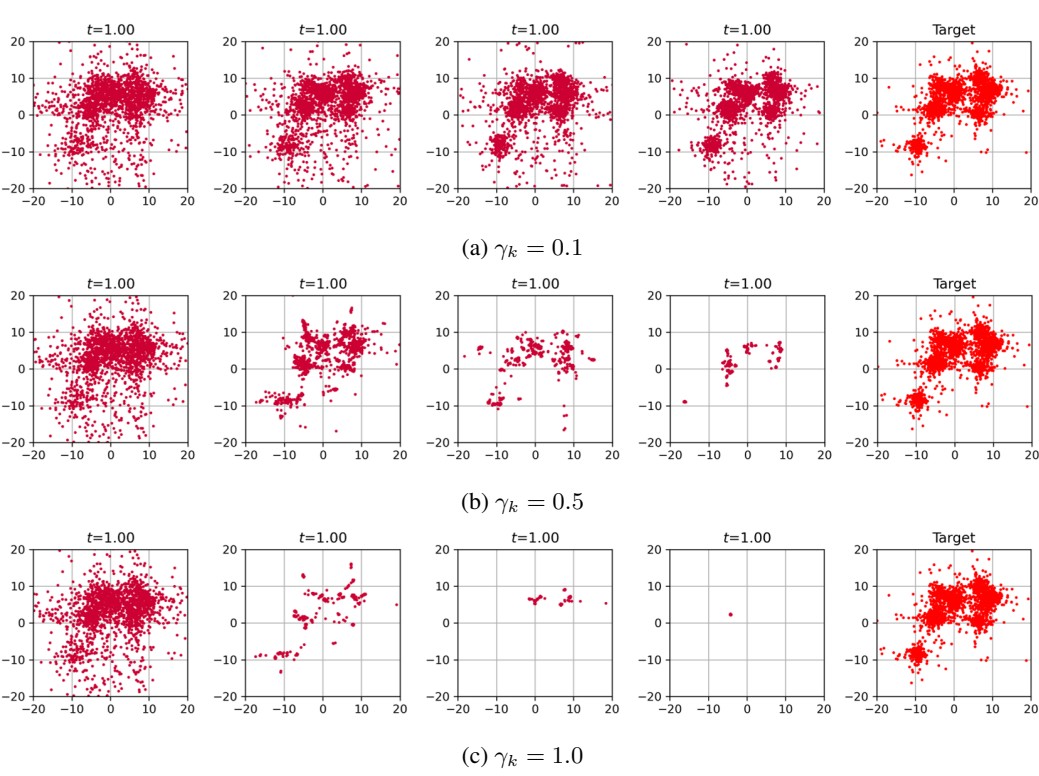

(a) $\gamma_k = 0.1$

(b) $\gamma_k = 0.5$

(c) $\gamma_k = 1.0$

Figure 5: Ablation studies on fixed $\gamma_k$ for all stages $k$ on MoS benchmark. We fix $\gamma_k = \frac{\eta_k}{1+\eta_k}$ to a constant for all stages $k$ and visualize the first four stages ($k = 1, 2, 3, 4$; left to right). Larger $\gamma_k$, i.e., weak proximal regularization, leads to rapid mode collapse whereas smaller $\gamma_k$ preserves multimodal coverage. These results are consistent with the analysis in Sec. 3.1.

### D.3 EFFECT OF PROXIMAL POINT METHOD

In this section, we study how the proximal term influences training and sampling quality. We first compare our *proximal WDCE* against the non-proximal WDCE baseline under identical settings. We then perform ablations over the proximal step size $\{\eta_k\}$ and its scheduling policy, reporting metrics across epochs (e.g., Sinkhorn and MMD) to quantify convergence and mode coverage.

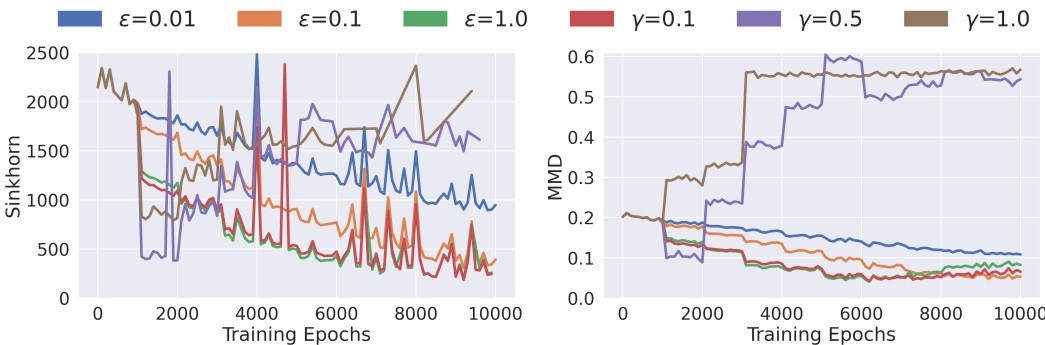

Figure 6: **Ablation on proximal step size $\eta_k$ and the choice of the scheduler on MoS.** We evaluate Sinkhorn ($\downarrow$) and MMD ($\downarrow$) across training epochs for multiple choices of proximal step size $\eta_k$ and scheduling policy. $\gamma$ denotes $\gamma_k := \frac{\eta_k}{1+\eta_k}$ over stage $k$. The legend entry "$\gamma = \text{const}$" denotes runs with a fixed $\gamma_k$ for all stages $k$. Note that $0 < \gamma_k \leq 1$; larger $\gamma_k$ weakens the proximal effect and approaches the non-proximal WDCE variant, whereas smaller $\gamma_k$ imposes stronger regularization.

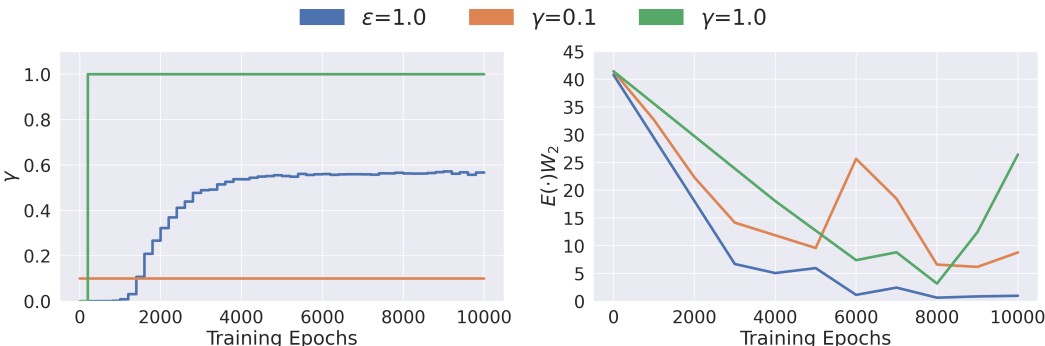

Figure 7: **Ablation on proximal step size $\eta_k$ and the choice of the scheduler on LJ-13.** We monitor $\gamma$ and energy 2-Wasserstein distance ($E(\cdot)\mathcal{W}_2(\downarrow)$) across training epochs for multiple choices of proximal step size $\eta_k$ and scheduling policy. $\gamma$ denotes $\gamma_k := \frac{\eta_k}{1+\eta_k}$ over stage $k$. The legend entry "$\gamma = \text{const}$" denotes runs with a fixed $\gamma_k$ for all stages $k$. Note that $0 < \gamma_k \leq 1$; larger $\gamma_k$ weakens the proximal effect and approaches the non-proximal WDCE variant, whereas smaller $\gamma_k$ imposes stronger regularization.

**Baseline vs. Proximal** We study the step size by using a fixed schedule, i.e., constant $\eta_k$ across stages. Define $\gamma_k := \frac{\eta_k}{1+\eta_k}$, which appears as the tempering exponent on the path weights in (8). The original WDCE is recovered at $\gamma_k = 1$ ($\eta_k = \infty$). We visualize generated samples for the first four stages ($k = 1, 2, 3, 4$) at $\gamma_k \in \{0.1, 0.5, 1.0\}$. As shown in Fig. 5, when $\gamma_k$ is large (0.5 or 1.0), the sampler quickly collapses onto partial modes in the initial stages. In contrast, $\gamma_k = 0.1$ preserves all modes across stages, indicating that stronger proximal regularization early in training helps prevent mode collapse and yields steadier progression toward the target.

**Effect of $\epsilon$** Now we analyze the effect of $\epsilon$. In using adaptive scheduler, $\epsilon$ is the most important hyperparameter since it determines proximal step size $\eta_k$ for each stage $k$. Decreasing $\epsilon$ tightens the trust region (smaller $\eta_k$, smaller $\gamma_k$), which tempers weights, reduces gradient variance, and improves stability; however, it results the slower convergence. Increasing $\epsilon$ relaxes the constraint (larger $\eta_k$, $\gamma_k \approx 1$), approaching the WDCE baseline with faster but potentially unstable steps (e.g., weight collapse).

On the MoS benchmark (Fig. 6), a very small radius ($\epsilon = 0.01$) is robust but converges too slowly to reach the best scores within the allotted epochs. A moderate radius ($\epsilon = 0.1$) attains steady, monotone improvements, while a large radius ($\epsilon = 1$) converges quickly at first but exhibits mild degradation in the final epochs. Overall, the range $0.1 \leq \epsilon \leq 1$ offers a favorable stability-speed trade-off.

**Effect of Adaptive Scheduling** As shown in Fig. 7, under the adaptive rule, the selected $\gamma_k :=$ $\frac{\eta_k}{1+\eta_k}$ tends to increase over stages (Fig. 7): small, conservative steps are chosen when the sampler is far from the target; larger steps become feasible as the model's support aligns with the target and more trajectories carry useful signal. This aligns to our intuition. In early stages, only a few samples lie near the target, producing a highly skewed distribution of pathwise weights in which a small set of trajectories dominates the loss; a smaller $\eta_k$ (or $\gamma_k$) tempers these weights, stabilizes gradients, and prevents premature mode concentration. On LJ-13, the adaptive scheduler consistently outperforms fixed-$\gamma$ schedule.

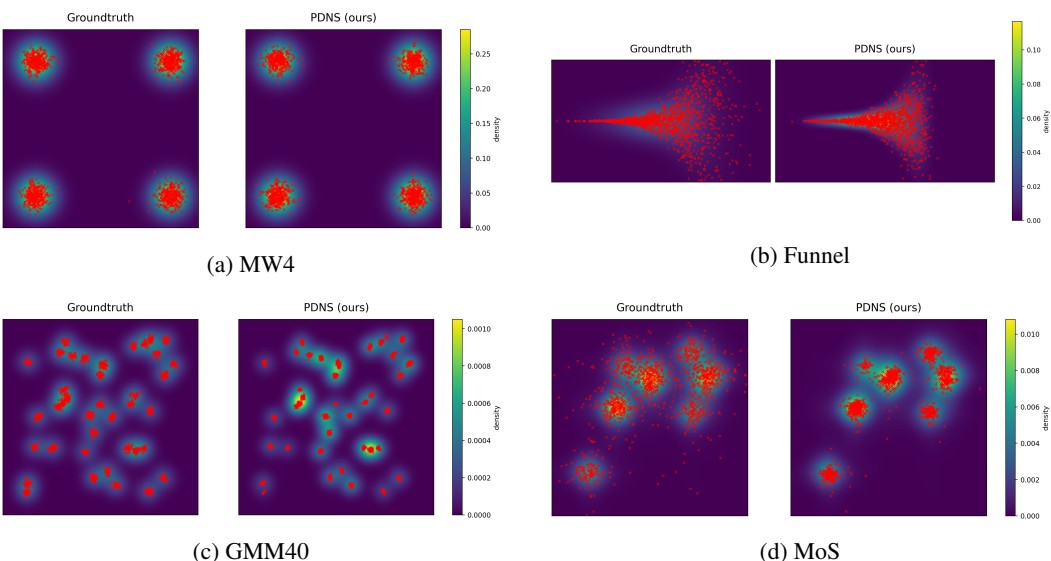

(a) MW4

(b) Funnel

(c) GMM40

(d) MoS

Figure 8: 2D kernel density estimates of ground-truth vs. generated samples, projected onto first two coordinates, for four synthetic benchmarks: (a) MW54, (b) Funnel, (c) GMM40, (d) MoS. The samples are shown as red dots.

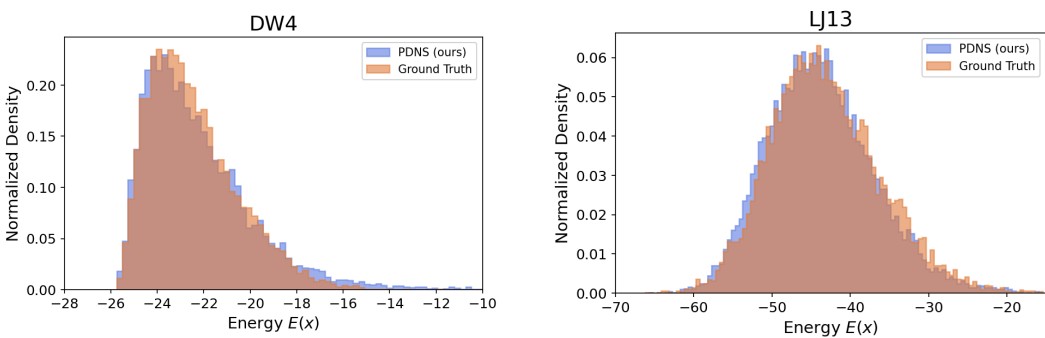

Figure 9: Energy histograms for DW-4 and LJ-13. PDNS produces energy distributions that closely match those of the ground-truth samples.

## D.4 ADDITIONAL QUALITATIVE RESULTS

**Synthetic Energy** We present qualitative results of our learned continuous diffusion samplers. First, we visualize 2D kernel density estimation (KDE) plot for 4 synthetic benchmarks (MW54,Funnel, GMM40, MoS) projected onto first two coordinates. We use 10000 samples to estimate KDE. The samplers cover all modes on MW-54, GMM40, and MoS, and capture the steep, narrow basin in Funnel. We further compare energy histograms under the true potentials for DW-4 and LJ-13 (Fig. 9); the distributions of $E(x)$ for PDNS closely match those of the ground-truth samples, providing qualitative support for the quantitative results in Tab. 3.

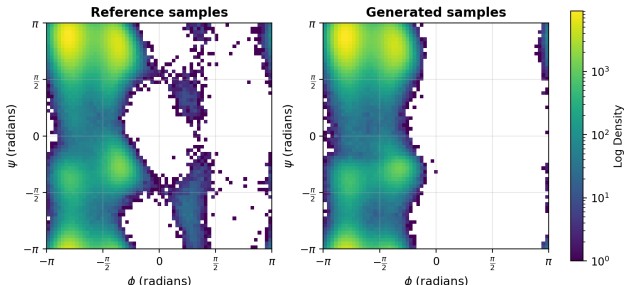 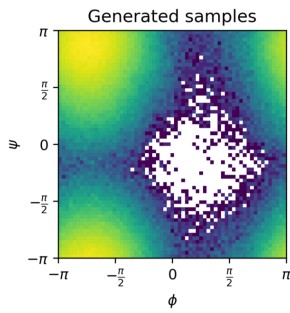

Figure 10: Ramachandran plot for AD between ground-truth and PDNS.

Figure 11: Ramachandran plot for $\mathbb{P}^0$.

**Alanine Dipeptide**    For the AD task, the Ramachandran plot in Fig. 10 shows that our model misses a few modes with extremely low probability mass. This mode collapse originates from the initial reference samples produced by the initial path measure $\mathbb{P}^0$, which already suffers from mode collapse (see Fig. 11). Because the proximal updates refine $\mathbb{P}^0$ in a progressive manner, modes absent in $\mathbb{P}^0$ may not be recovered during training. Note that Fig. 11 plots all samples with energy $E(x) < 1000$ obtained from $10^6$ terminal samples drawn from $\mathbb{P}^0$, highlighting the missing low-probability regions. We leave the exploration of improved initialization strategies for constructing a better $\mathbb{P}^0$ as an future work.

# E    EXPERIMENTAL DETAILS AND FURTHER RESULTS FOR LEARNING DISCRETE TARGET DISTRIBUTIONS

## E.1    FURTHER EXPLANATION ON THE TARGET DISTRIBUTIONS

**Lattice Ising and Potts Models**    We consider Ising and Potts models defined on a square lattice with $L$ sites per dimension, $\Lambda = \{1, ..., L\}^2$. The state space is $\mathcal{X}_0 = \{\pm 1\}^\Lambda$ for the Ising model and $\mathcal{X}_0 = \{1, ..., q\}^\Lambda$ for the Potts model with $q$ spin states. Given an interaction parameter $J \in \mathbb{R}$, the potential of any configuration $x \in \mathcal{X}_0$ is given by[4]

$$\text{Ising: } E(x) = -J\sum_{i \sim j} x^i x^j; \quad \text{Potts: } E(x) = -J\sum_{i \sim j} 1_{x^i = x^j}, \tag{66}$$

where $i \sim j$ means that $i, j \in \Lambda$ are adjacent on the lattice. For simplicity, we impose periodic boundary conditions in both the horizontal and vertical directions. With inverse temperature $\beta > 0$, the target distribution is the Gibbs measure $\pi(x) \propto e^{-\beta E(x)}$.

**Combinatorial Optimization**    The energy of the target distribution for solving the maximum cut problem is defined as $E(x) = -\sum_{i \sim j} \frac{1 - \sigma_i \sigma_j}{2}$, where $\sigma_i = 2x_i - 1$.

Note that $E$ and $\pi$ are now graph-dependent, but we omit this in the notation for simplicity.

## E.2    IMPLEMENTATION DETAILS

### E.2.1    MODEL BACKBONE

**Lattice Ising and Potts Models**    Following the implementation in Zhu et al. (2025), we use vision transformers (ViT, Dosovitskiy et al. (2021)) as the backbone for the discrete diffusion model. Specifically, we adopt the DeiT (Data-efficient image Transformers) framework (Touvron et al., 2021) with 2-dimensional rotary position embedding (Heo et al., 2025), which effectively captures the 2-dimensional spatial structure. For learning both the Ising model with $L = 24$ and the Potts model with $L = 16$ and $q = 4$, we use 4 blocks with a 128-dimensional embedding space and 4 attention heads, and the whole model contains 829k parameters, which is larger than the model used in Zhu et al. (2025) due to more challenging target distributions here.

---

[4]We do not consider the external field for simplicity.

**Combinatorial Optimization** We leverage a variant of diffusion transformer (DiT, Peebles & Xie (2023)) known as the Graph DiT (Liu et al., 2024), with a 64-dimensional hidden space, 6 blocks and 8 heads, and the total number of parameters is 483k.

Suppose the graphs in the data set have at most $D$ nodes. The score network $s_\theta$ takes the following three components as input: (1) $x \in \mathcal{X} = \{0, 1, 2 = \mathbf{M}\}^D$, (2) $\Phi(G) \in \mathbb{R}^{D \times E}$ the feature of the graph $G$ obtained from **adjacency spectral embedding**, and (3) $v(G) \in \{0, 1\}^D$ an indicator of whether a node is valid in graph or just dummy (padding) node. $\Phi$ is constructed as follows: suppose the graph $G$, after padding dummy nodes, has $D$ nodes. Let $A(G) \in \mathbb{R}^{D \times D}$ be the adjacency matrix of $G$, whose eigenvalue decomposition is written as $A(G) = U \Lambda U^{\mathrm{T}}$. Then given the dimension of graph feature $E \in [1, D]$, $\Phi(G) := U_{[:,1:E]} \Lambda_{[1:E,1:E]}^{\frac{1}{2}} \in \mathbb{R}^{D \times E}$. It is easy to verify that adding padding nodes does not change the embedding, except for appending lines of zeros. Through all of our experiments, we choose $E = D/2$ by default.

### E.2.2 TRAINING HYPERPARAMETERS

**Lattice Ising and Potts Models** For these distributions, we choose the batch size as 256, and use the AdamW optimizer (Loshchilov & Hutter, 2019) with a constant learning rate of 0.001. We always use exponential moving average (EMA) to stabilize the training, with a decay rate of 0.9999. In computing the loss, we use 4 replicates per buffered sample for the Ising model and 8 for the Potts model due to the memory constraint, and resample the buffer every 10 gradient updates with the buffer size $\mathcal{B} = 512$. All experiments are trained on an NVIDIA H100 80GB HBM3.

For learning both models at $\beta_{\mathrm{critical}}$, we use 25 outer loops, each with 4000 steps of gradient update. In the first 20 outer loops, the $\lambda_k$ decreases linearly from 1 to 0; for the last five outer loops, $\lambda_k$ is set to 0 (i.e., the step size $\eta_k = \infty$ so that we refine the final target distribution). To learn the models at $\beta_{\mathrm{low}}$, we start from the best checkpoint at $\beta_{\mathrm{critical}}$ and directly use infinite step size.

**Combinatorial Optimization** For the maximum cut problem, we choose the inverse temperature of the target distribution as $\beta = 5$, and use three outer loops with $\lambda_1 = 0.8, \lambda_2 = 0.4, \lambda_3 = 0$. For each outer loop in Alg. 1, we train the model for 20 inner epochs. In each epoch, we first randomly sample 32 graphs from the training set, and for each graph, we generate 16 samples along with their weights as buffer. Then, we iterate over the samples in the buffer one by one for 10 rounds. Each time, the samples that correspond to the same graph along with their weights are used to compute the loss. All experiments are trained on an NVIDIA RTX A6000.

### E.2.3 GENERATING BASELINE AND GROUND-TRUTH SAMPLES

**Lattice Ising and Potts Models** For learning-based baseline, we mainly compare with **LEAPS** (Holderrieth et al., 2025). We use the same model backbone as introduced in Holderrieth et al. (2025). For a fair comparison, we adjust the hyperparameters to ensure the models have comparable number of parameters as above: we choose kernel sizes $[3, 5, 7, 9, 11, 13, 15]$ and 32 channels. For learning Ising models, the number of input channels is 1, making the total number of parameters 919k, while for learning Potts models with $q = 4$, the number of input channels is 20, making the total number of parameters 928k. For each inverse temperature, we train LEAPS for up to 100k steps for each temperature, which is comparable to ours.

We also include two MCMC baselines. First, we use the **Metropolis-Hastings (MH)** algorithm with single-site updates. For both Ising and Potts models, and among all temperatures, we use a batch size 1024, and warm up the algorithm with $2^{20} = 1048576$ burn-in iterations. After that, we collect the samples every $2^{16} = 65536$ steps to ensure sufficient mixing, and collect for 64 rounds to gather a total of $2^{16} = 65536$ samples. Moreover, we run the **SW algorithm** on all target distributions to generate examples accurately distributed as the ground truth. We use a batch size of 256, and warm up the algorithm with $2^{13} = 8192$ burn-in iterations. After that, we collect samples every 128 steps to ensure sufficient mixing of the chain, and collect for 256 rounds to gather a total of $2^{16} = 65536$ samples.

For evaluation, we draw $2^{12} = 4096$ samples from each trained model along with their weights to compute the ESS (see Sec. E for explanations for the computation of ESS). We also use the same number of samples from the MH algorithm. To obtain a precise estimate of the ground truth, we use all 65536 samples from SW to compute the magnetization and 2-point correlation.

We refer readers to Zhu et al. (2025, App. D.3 and E.2) for the definitions of the evaluation metrics (magnetization and 2-point correlation) for Ising and Potts models, respectively. In particular, Fig. 4 plots the average 2-point correlation $\frac{1}{2}(C^{\mathrm{row}}(k, k+r) + C^{\mathrm{col}}(k, k+r))$ therein.

**Combinatorial Optimization**    For each type of graph, we generate 1024 graphs as the training set, and another 32 graphs as the test set. We solve the problems by Gurobi (Gurobi Optimization, LLC, 2025) and take the output as the ground truth.

### E.3    ADDITIONAL RESULTS

**Monitoring Training Procedure of PDNS**    In Fig. 12, we demonstrate an example of the PNDS learning procedure for the Potts model with $q = 4$ states on a $16 \times 16$ lattice, at $\beta_{\mathrm{critical}} = 1.0986$. In the $k$-th outer loop, we fit the target path measure $\mathbb{P}^k$.

We now illustrate the computation of the effective sample size (ESS). For i.i.d. trajectories $X^{(1:M)} \sim \mathbb{P}^\theta$, associate $X^{(j)}$ with a local weight $w_j \propto \frac{\mathrm{d}\mathbb{P}^k}{\mathrm{d}\mathbb{P}^\theta}(X^{(j)})$ computed through (15) and (16), and normalize them such that $\sum_{j=1}^M w_j = 1$. Associate $X^{(j)}$ with a global weight $w'_j \propto \frac{\mathrm{d}\mathbb{P}^*}{\mathrm{d}\mathbb{P}^\theta}(X^{(j)}) \propto \frac{\mathrm{d}\mathbb{P}^{\mathrm{ref}}}{\mathrm{d}\mathbb{P}^\theta}(X^{(j)})\mathrm{e}^{r(X_T^{(j)})}$ computed through (16), and also normalize them such that $\sum_{j=1}^M w'_j = 1$.

Then, the local ESS w.r.t. $\mathbb{P}^k$ is defined by $\left(M \sum_{j=1}^M w_j^2\right)^{-1}$, and the global ESS w.r.t. $\mathbb{P}^*$ is defined by $\left(M \sum_{j=1}^M {w'_j}^2\right)^{-1}$. Both values are in $\left[\frac{1}{M}, 1\right]$, with a higher value typically indicating a better fit between two path measures.

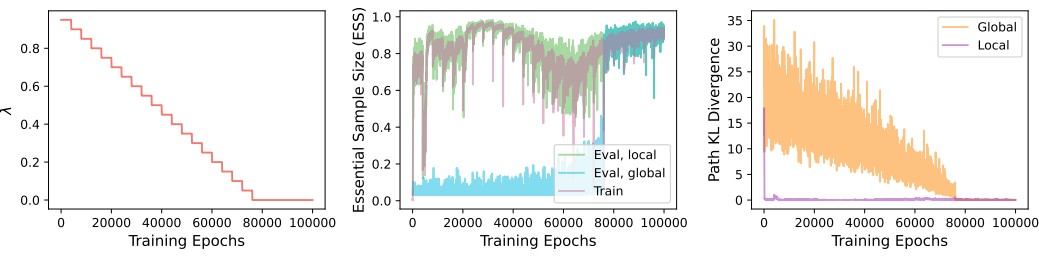

Figure 12: An example of PNDS training for Potts model with $q = 4$ states on a $16 \times 16$ lattice, at $\beta_{\mathrm{critical}} = 1.0986$. **Left:** the schedule of $\lambda_k$'s. We use 25 outer loops with 4000 steps per outer loop. In the initial 20 outer loops, we choose step size $\eta_k$'s such that $\lambda_k$'s decrease linearly from 1 to 0. The last 5 outer loops use $\eta_k = \infty, \lambda_k = 0$ to refine the final target. **Middle:** the ESS for training and evaluation during training. **Right:** the KL divergence between path measures during training. In the middle and right plots, "global" means with respect to $\mathbb{P}^*$ and "local" means with respect to $\mathbb{P}^k$.

**Ablation Study on Scheduler**    Throughout this part, the target distribution we consider is the lattice Ising model with $L = 8$, $J = 1$, and $\beta_{\mathrm{low}} = 0.6$, which shares the same physical property as the Ising mode on larger lattices presented in Sec. 5.2.

In Fig. 13, we test several predefined schedules for $\eta_k$ and $\lambda_k$: three of them are with linearly decreasing $\lambda_k$ and each stage $1 \le k \le K$ uses the same number of training epochs, and the remaining three are with constant $\gamma_k = \frac{\eta_k}{1+\eta_k}$, and the number stages $K$ is chosen such that $\lambda_K$ is around 0.01 (the final $\gamma_K$ is set to be one). We fix the same number of training epochs (10000). We observe that all schedulers tested are able to learn the correct target distribution, while schedulers with constant $\gamma_k$ shows faster convergence in terms of the metrics due to its faster decrease in the corresponding $\lambda_k$ at initial stages.

In Fig. 14, we test the effect of $\epsilon$ on the adaptive schedules, where we recall that the next target $\mathbb{P}^{k^*}$ is chosen such that $\widehat{\mathrm{KL}}(\mathbb{P}_{k-1}^\theta \| \mathbb{P}^{k^*}) \le \epsilon$. We choose the values of $\epsilon$ ranging from 0.05 to 5. Here, for each subproblem $k - 1$, we run at least 100 gradient updates to the model, and test whether the local ESS w.r.t. $\mathbb{P}^{k-1}$ (defined at the previous part) is $\ge 0.95$. If so, we adaptively select the next $\eta_k$ through solving the optimization problem (19). We found that this adaptive scheduler is quite robust

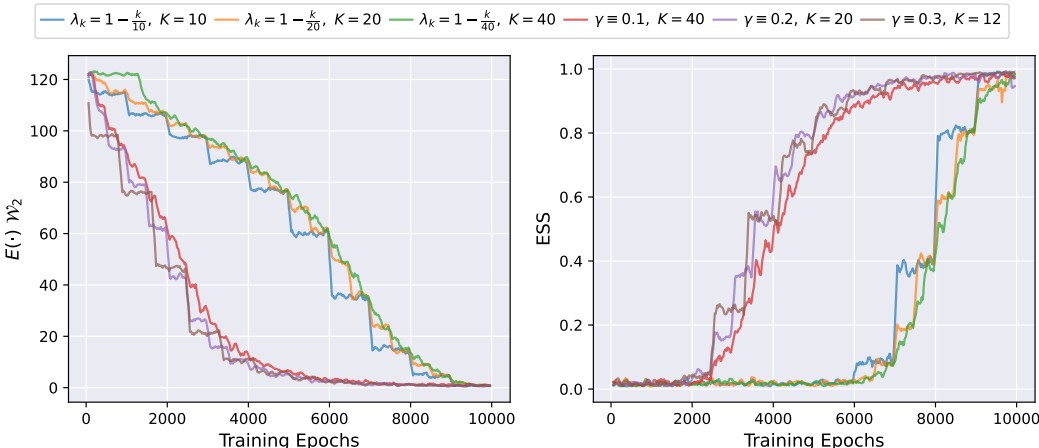

Figure 13: Ablation on the choice of the *prefixed* schedulers on learning lattice Ising model with $L = 8$, $J = 1$, and $\beta_{\text{low}} = 0.6$. **Left**: the energy 2-Wasserstein distance $E(\cdot)\ \mathcal{W}_2$ between the generated and ground-truth samples; **Right**: the effective sample size (ESS) of the samples from $\mathbb{P}^\theta$ with respect to the optimal path measure $\mathbb{P}^*$.

to the choice of $\epsilon$, and in all choices tested, the total numbers of training steps are smaller than the naïvely chosen total number of training steps in the prefixed schedulers above.

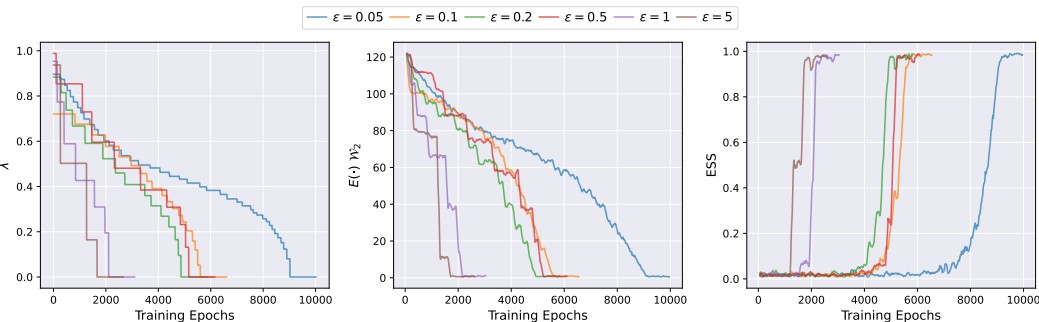

Figure 14: Ablation on the choice of the *adaptive* schedulers on learning lattice Ising model with $L = 8$, $J = 1$, and $\beta_{\text{low}} = 0.6$. **Left**: the adaptive selected $\lambda_k$'s; **Middle**: the energy 2-Wasserstein distance $E(\cdot)\ \mathcal{W}_2$ between the generated and ground-truth samples; **Right**: the effective sample size (ESS) of the samples from $\mathbb{P}^\theta$ with respect to the optimal path measure $\mathbb{P}^*$.

**Additional Qualitative Results**    For lattice Ising and Potts models, we visualize the generated samples under each target distribution in Figs. 15 to 18, where we observe that our PNDS method learns the correct target distribution. We also compare the energy histograms in Figs. 19 and 20. The distributions of $E(x)$ (defined in (66)) for samples from PDNS closely match those of the ground-truth samples generated by the SW algorithm, indicating that PDNS is capable of learning the correct target distribution. We use in total $4096$ samples from both methods in each distribution.

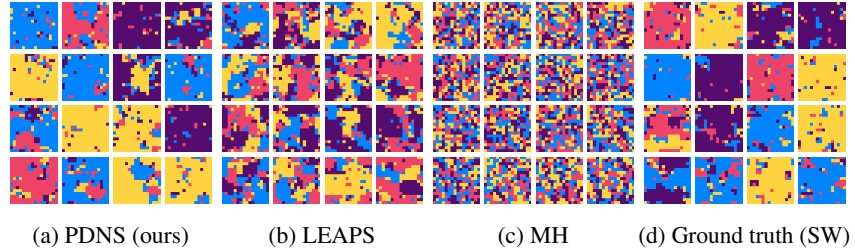

(a) PDNS (ours)      (b) LEAPS      (c) MH      (d) Ground truth (SW)

Figure 15: Visualization of non-cherry-picked samples from the learned $16 \times 16$ Potts model with $J = 1$, $q = 4$, and $\beta_{\text{critical}} = 1.0986$. (a) MDNS. (b) LEAPS. (c) MH. (d) Ground truth (simulated with SW algorithm).

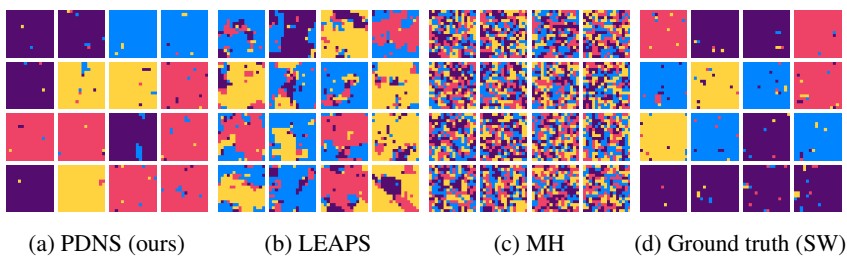

(a) PDNS (ours)      (b) LEAPS      (c) MH      (d) Ground truth (SW)

Figure 16: Visualization of non-cherry-picked samples from the learned $16 \times 16$ Potts model with $J = 1$, $q = 4$, and $\beta_{\text{low}} = 1.3$. (a) MDNS. (b) LEAPS. (c) MH. (d) Ground truth (simulated with SW algorithm).

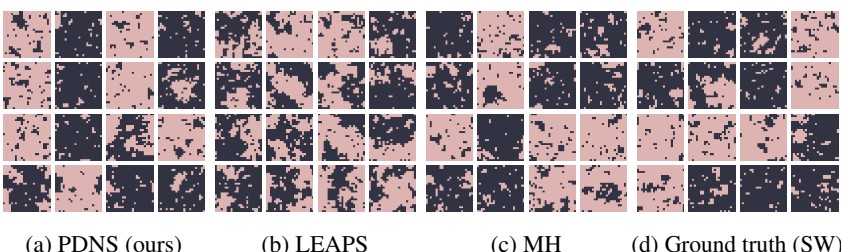

(a) PDNS (ours)      (b) LEAPS      (c) MH      (d) Ground truth (SW)

Figure 17: Visualization of non-cherry-picked samples from the learned $24 \times 24$ Ising model with $J = 1$ and $\beta_{\text{critical}} = 0.4407$. (a) MDNS. (b) LEAPS. (c) MH. (d) Ground truth (simulated with SW algorithm).

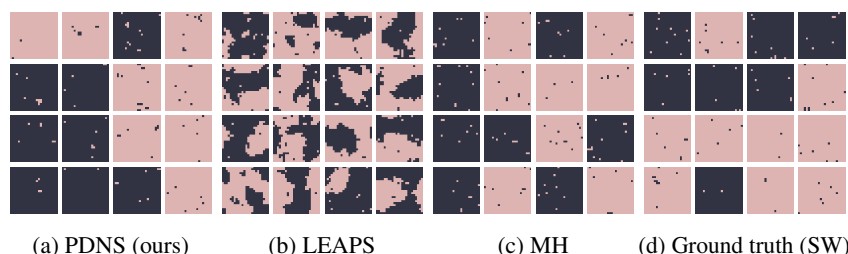

(a) PDNS (ours)      (b) LEAPS      (c) MH      (d) Ground truth (SW)

Figure 18: Visualization of non-cherry-picked samples from the learned $24 \times 24$ Ising model with $J = 1$ and $\beta_{\text{low}} = 0.6$. (a) MDNS. (b) LEAPS. (c) MH. (d) Ground truth (simulated with SW algorithm).

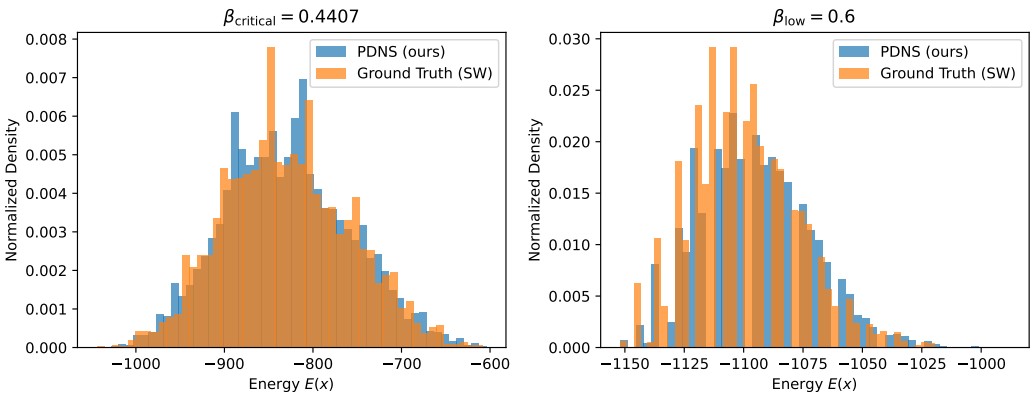

Figure 19: Energy histograms for Ising models with $L = 24$ and $J = 1$.

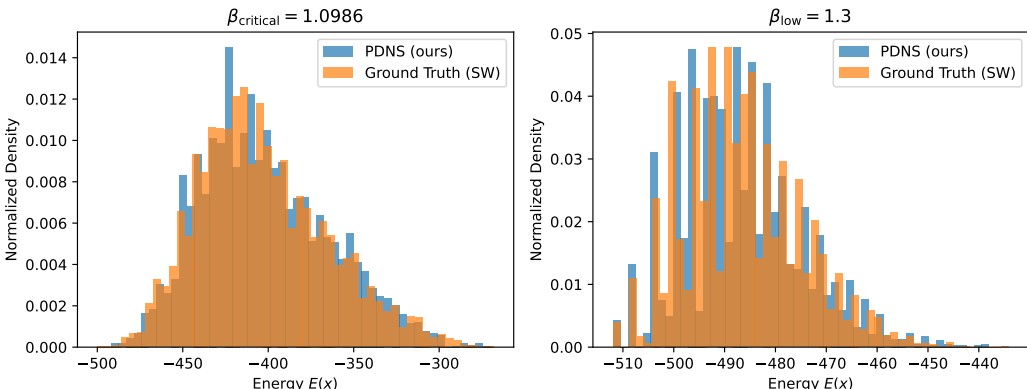

Figure 20: Energy histograms for Potts models with $L = 16$, $J = 1$, and $q = 4$.

