# OpenReview forum: "Proximal Diffusion Neural Sampler"
_ICLR.cc/2026/Conference — ICLR 2026 Poster_

### Official Review · Reviewer_h5wm · 2025-10-15

**Soundness:** 4
**Presentation:** 3
**Contribution:** 3
**Rating:** 6
**Confidence:** 3

**Summary:**

The work proposes the “Proximal Diffusion Neural Sampler”. The authors consider the setting of sampling from distribution with given unnormalized densities on both discrete and continuous state spaces. Rather than solving the SOC/sampling problem in one step, they propose an iterative procedure where an intermediate set of path measures are approximated, starting with a simple path measure and a final target path measure. It is shown that these intermediate path measure correspond to a geometric annealing in path space (proposition 3.1). Every step of the optimization problem is solved with proximal WDCE, a variant of WDCE designed for the proximal setup of this work.

**Strengths:**

- Novel idea that is well-motivated with a technically good presentation.
- A nice and simple unifying framework of existing neural sampler methods. They consider both discrete and continuous setting in one (instead of writing two separate papers, one general principle is presented).
- Great illustration of mode collapse (section 3.1) motivating the work.
- Experiments show improvements that illustrate the theoretical innovations.

**Weaknesses:**

- For a less-experienced reader, the paper would potentially be hard to follow. I recommend including more sentences explaining intuitions and motivations at intermediate paragraphs.
- The experiments are only on simple distributions. While this unfortunately common in this line of literature, a more complex setting of higher-dimensional distributions would have been more convincing.

**Questions:**

- Line 72: “path space, which contains all functions from [0, T] to X”. A path measure is only defined for a subset of functions usually (there is a measurable space, etc.)
- The ESS for the LEAPS algorithm reported here is significantly lower than report in their work. Comparing the setting, you have slightly changed the settings (slightly different grid size) or q=4 instead of q=3 for the Potts model. Why did you change this setting? It would be more compelling if you compare to exactly the same parameters than their work.

---

> ### Author Response · Authors · 2025-11-21
>
> We sincerely thank the reviewer for the positive feedback and constructive comments. We are glad that the reviewer found our idea novel and the paper well-motivated with a clear presentation. Below, we address the comments point-by-point.
>
> ---
>
> > **Weakness 1**: Including more sentences explaining intuitions and motivations at intermediate paragraphs
>
> We appreciate the reviewer's suggestion to improve the clarity and readability of our presentation for a broader audience. While Sec. 3.1 and the opening paragraph of Sec. 3.2 already provide the intuition behind the limitations of the original WDCE method and motivation of decomposing the problem into simpler subproblems, we agree that certain parts of the presentations would benefit from additional explanation.
>
> - In particular, we recognize that the general (non-proximal) optimization problem in (3) may be challenging to interpret on first reading. To address this, we have added additional intuitive explanation in the main text, including the discussion on the memoryless reference dynamics.
>
> - We have also added further explanation of the proximal optimization problem in (7), clarifying why the proximal point method provides significant advantages in line 221-228.
>
> - Moreover, because some of our notation may have been hard to interpret, we have added additional explanation after Prop. 3.1 and included Tab. 1 for summarizing the path-measure–related notations.
>
> We hope all these modifications can help the general audience better understand our work.
>
> ---
>
> > **Weakness 2**: More complex target distributions
>
> Thanks for this suggestion. Being capable of handling more complex high-dimensional target distributions is always a goal for training diffusion samplers. The target distributions we considered such as $24\times24$ Ising models and $16\times16$ Potts model with $q=4$ states are already the largest among other works on neural samplers. For continuous target distributions, we have also included new results on alanine dipeptide molecular generation task in Sec. 5, and we show that PDNS reaches state-of-the-art performance.
>
>
> ---
>
> > **Question 1**: The definition of path measure should be modified.
>
> Thanks for pointing out this important point. We completely agree that the path space has a more rigorous definition. For instance, for continuous SDEs, the path space is usually defined as the space of continuous functions from $[0,T]$ to $\mathcal{X}$ with certain regularity conditions (e.g., with bounded quadratic variation); for CTMCs, the path space is the space of piecewise constant and càdlàg (right-continuous with left limits) functions. Here, we just say the paths are in a subset of the space of all functions from $[0,T]$ to $\mathcal{X}$, which is easier to understand for the general audience. We have added a footnote in the revised manuscript to clarify this point.
>
> ---
>
> > **Question 2**: ESS for the LEAPS algorithm
>
>
> Thanks for raising this issue. For the Potts model, we chose a slightly larger size and a larger number of configurations $q$ ($4$ rather than $3$). The purpose of doing so is to ensure that the learning target is sufficiently more challenging than the one tested in previous works, as the number of possible states increases $(4/3)^{16\times16}=9.6\times10^{31}$ times. For fairness of comparison, we have adjusted the size of neural networks: PDNS' network has 829k parameters while LEAPS' network has 928k parameters. Also, we have used significantly more compute for training LEAPS until its performance plateaued. We refer readers to App. E.2 for further implementation details.
>
> ---
>
> Finally, we sincerely thank the reviewer again for the valuable feedback, and we hope that our responses and the revised manuscript address the raised concerns. Please feel free to reach out with any further questions or comments.

---

> > ### Comment · Reviewer_h5wm · 2025-11-22
> >
> > Thank you for addressing my concerns in your revised manuscript and providing answers on my questions. I read the revised version and the new version is easier to understand. I also appreciated the additional experiments. I would strongly encourage the authors to scale up their method to real systems in the future (alanine dipeptide is a toy systems for any computational chemist). However, I acknowledge that the primary contribution of this work lies in a proof-of-concept of a novel theoretical framework. I therefore increase my score.

---

> > > ### Author Response · Authors · 2025-11-22
> > > **Thank you for increasing the score!**
> > >
> > > We would like to express our sincere gratitude to the reviewer for raising the score. We are glad to hear that our modified manuscript has improved readability, and we will continue pursuing for scalable neural sampling methods targeting challenging real-world problems in the future.

---

### Official Review · Reviewer_prWn · 2025-10-30

**Soundness:** 3
**Presentation:** 3
**Contribution:** 3
**Rating:** 8
**Confidence:** 4

**Summary:**

This paper proposed Proximal Diffusion Neural Sampler (PDNS) to learn a neural sampler for sampling from unnormalised density functions.   Both continuous and discrete formulation is proposed and verified in this paper.
The paper argus that the reason behind model collpasing is that the sampling task is too far from the prior and hence PDNS breaks down the entire problem with proximal point method, progressively driving the neural sampler towards the target.
This proposed method enriches the family of neural samplers.

**Strengths:**

The paper is clearly written and easy to follow. It is complete and well structured. It covers both discrete and continuous domains and presents results for each. The proposed algorithm is interesting and offers insights into mode collapse. The empirical evaluation indicates that the approach is promising.

**Weaknesses:**

While the paper is motivated by mitigating mode collapse, the experimental section does not convincingly demonstrate this. Several benchmarks (e.g., LEAPS) do not appear to exhibit severe collapse. Do you have any explanation to this?


On line 780:
> Blessing et al.  (2025) couple CE training... In contrast,PDNS utilize
 proximal algorithm with WDCE bjective,which can be applied uniformly to continuous-time
 and discrete-time diffusions,and is designed to be memory-efficient while retaining the benefits of
 denoising or score-matching objectives.

The mentioned algorithm has quite similar formulation and has strong connections. Blessing et al.  (2025)  works on continuous. But to my knowledge, it should also be able to work on discrete domain. Is this the main difference between PDNS and their approaches? Or is my understanding wrong?

**Questions:**

See my weakness

---

> ### Author Response · Authors · 2025-11-21
>
> We sincerely thank the reviewer for the positive feedback and valuable comments. We are delighted that the reviewer found our paper well-structured and interesting. Please find our responses to the comments below.
>
> ---
> > **Weakness 1**: Explanation to how to mitigate the mode collapse
>
> Thanks for pointing out this important point. As we mentioned in Sec. 3, the mode collapse issue arises when directly training a sampler from scratch to fit the untempered target distribution, especially when the target distribution is complex with multiple modes. In our experiments, the baseline method LEAPS [1] does suffer from this issue, which can be seen from the wrong 2-point correlation computed on the generated samples in Fig. 4, and also from the sample visualization in Figs. 15 - 18. All these signs suggest that the sampler may fail to learn to generate the most plausible modes of the target distribution.
>
> ---
>
> > **Weakness 2**: Difference to Blessing et al. (2025)
>
> We appreciate the reviewer giving us the opportunity to clarify the differences between our work and the recent concurrent method by Blessing et al. [2]. While both approaches incorporate proximal or trust-region ideas to stabilize diffusion-based samplers, our method differs in several key aspects:
>
> - [2] use the log variance (LV) loss and the stochastic optimal control matching (SOCM) loss, both of which require storing full trajectories during training. In contrast, our method minimizes the weighted denoising cross-entropy (WDCE) loss, which eliminates the need for trajectory storage.
>
> - As discussed in MDNS [3], trajectory-based losses (e.g. LV) may face challenges when scaling to high-dimensional discrete sampling tasks. Our approach mitigates this by storing only samples at the terminal time step and applying importance reweighting, which simplifies memory usage and supports scalability in both continuous and discrete settings.
>
> - [2] only considers continuous target distributions. In contrast, our method demonstrates applicability in both continuous and discrete diffusion-based sampling tasks with empirical results.
>
> - Our method adopts a more general formulation by considering both $\mathbb{P}^{\theta^ * _ k}$ and $\mathbb{P}^k$ ((8), (9) in Prop. 3.1 and Alg. 1) when defining the target path measure. [2] only focus on $\mathbb{P}^{\theta^ * _ k}$.
>
> In summary, while sharing high-level conceptual similarities, our method introduces a distinct objective function and broader formulation that supports both continuous and discrete settings within a unified framework.
>
> **References**:
>
> [1] Holderrieth et al. LEAPS: A discrete neural sampler via locally equivariant networks. ICML 2025.
>
> [2] Blessing et al. Trust Region Constrained Measure Transport in Path Space for Stochastic Optimal Control and Inference. NeurIPS 2025.
>
> [3] Zhu et al. MDNS: Masked Diffusion Neural Sampler via Stochastic Optimal Control. NeurIPS 2025.
>
> ---
>
> We thank the reviewer again for the positive evaluation of our work, and we hope that our responses and the revised manuscript address the concerns. We are happy to respond to any further questions or comments.

---

> > ### Comment · Reviewer_prWn · 2025-11-27
> >
> > Thank you for the reply, which addressed my questions. I will keep my score at 8. Congrats for the great work

---

### Official Review · Reviewer_Kemc · 2025-10-31

**Soundness:** 3
**Presentation:** 2
**Contribution:** 4
**Rating:** 8
**Confidence:** 3

**Summary:**

Proximal Diffusion Neural Sampler proposes a method an stochastic optimal control (SOC) sampler that learns a target distribution based on having an energy function alone. It is an extension of the current developments in this area, specifically by applying the proximal point method on the space of path measures. They provide a method that works for both discrete and continuous cases. The end result is an algorithm where one can do a variable amount of optimization problems that empirically (and theoretically) converge to the target distribution without mode collapse. Essentially, a computationally tractable method for "forward-KL" solving with SOC.

In general, the contribution is strong and fairly easy to follow given the complexity (although I suggest improvements below). It seems applicable with fairly good results.

**Strengths:**

# overall
- problem setting is well identified and known to be a practical issue
- background is well explained and covers both discrete and continuous cases
- In general, the key results from a hard topic are well presented... (The extensions from the paper need some notation work, in my opinion).
- The method introduces a scheme of iterates to help exploration, including a proposed scheduler that dynamically decides the number of iterations based on a bounded KL term. This is nice and well motivated.

# results
- results on non-particle continuous models are very good.
- results seem fairly good on particle models
- results are strong on the discrete models

**Weaknesses:**

# clarity in actual objective
- From lines 298 - 318 the paper does not make it extremely clear how the actual objective is computed. There are a number of identities used, in particular the right part of (15) that do not seem clearly motived by other equations in the text.
- Similarly, about clarity, You use so many version of $\mathbb{P}$ it becomes difficult to compare them. In particular the most confusing are the non-parametric ones: P^k, P^k*. P^k is a non-optimal, non-parametric path measure? It's not obvious why we even need this thing or how it is defined. Maybe it would be worth introducing alternative notation of P to help keep everything straight.
- ... Similarly, (16) refers to a ratio of path measure derivatives that does not appear in the discrete objective on 322-323. Can you please write it out?

# cost
- can you better explain the cost of training your model versus others? Is it k-times as expensive?

# unclear about some parts of method
- ASBS, for example, is extremely hyperparameter sensitive. How about your method?
- You use a memoryless path, correct? How does this interact with your optimization problem?
- Can you use a non-memoryless distribution, e.g. like ASBS does?

**Questions:**

# results
- can you please plot the energy histogram for the particle problems? It tells us much more than W2 and E() W2.
- can you please work on notation in the methods you are proposing a bit? I know they're all path measures...
- I know it's a lot to ask, but is there any molecular task you can apply this to? Alanine Dipeptide for example? Strong contender ASBS misses important modes, if your method gets them that would be great!
- Can you better quantify how much excess training cost is required to solve the k proximal problems? Does it really slow everything down by a lot?

---

> ### Author Response · Authors · 2025-11-21
>
> We sincerely thank the reviewer for the positive feedback and constructive comments. We are glad that the reviewer found our paper well-motivated and acknowledged the contributions of our work. Please find our point-by-point responses to the comments below.
>
> > **Weakness 1.1**: How the actual objective is computed is not clear enough
>
> We thank the reviewer for raising this point and helping us improve the clarity of our learning objective. We have modified this part for better readability.
>
> In our updated manuscript, the first line in (13) is the original DSBM loss with target path measure $\mathbb{P}^{k^ * }$ (which can be either $\mathbb{P}^{\theta _ k^ * }$ or $\mathbb{P}^k$ defined in Prop. 3.1), and in the second line we use importance sampling to rewrite the expectation under the detached path measure $\mathbb{P}^{\bar\theta _ {k-1}}$ obtained from the previous iteration. To compute the RN derivative $\frac{\mathrm{d}\mathbb{P}^{k^ * }}{\mathrm{d}\mathbb{P}^{\bar\theta _ {k-1}}}$, we consider two cases separately: for $\mathbb{P}^{k^ * }\gets\mathbb{P}^{\theta _ k^*}$ defined in (8), by combining this with (3), this leads to the identity on the left part of (15); for $\mathbb{P}^{k^ * }\gets\mathbb{P}^k$ defined in (9), we can again combine (9) with (3) to get the identity on the right part of (15). We have added more explanations below (13) in the revised manuscript to clarify this point.
>
> ---
>
> > **Weakness 1.2 & Question 2**: Clarity of notations for path measures.
>
> We appreciate the opportunity to clarify the confusion caused by the notation. We have added Tab. 1 under Prop. 3.1 to summarize the meaning of path measures and added more explanation there. We provide the clarification below:
>
> - $\mathbb{P}^{\theta^ * _ {k}}$: Given the approximate path measure $\mathbb{P}^{\theta _ {k-1}}$ obtained from the $(k-1)$-th proximal step, we define $\mathbb{P}^{\theta _ k^*}$ as the solution to the $k$-th proximal step (7) starting from $\mathbb{P}^{\theta _ {k-1}}$.  In other words, each $\mathbb{P}^{\theta^ * _ {k}}$ is derived by applying a proximal update starting from the current learned approximation of the path measure $\mathbb{P}^{\theta _ {k-1}}$.
>
> - $\mathbb{P}^{{k}}$: Given $\mathbb{P}^0$ as a reference measure and a sequence of proximal step sizes $\\{\eta _ k \\}^K _ {k=1}$, we define $\mathbb{P}^k$ as the *exact* optimal path measure for the $k$-th proximal step. A closed-form expression for $\mathbb{P}^k$ is in Prop. 3.2 (2.).
>
> - Our framework allows flexibility in choosing the target measure at each proximal step. Specifically, one can train the sampler to approximate either $\mathbb{P}^{\theta _ k^ * }$ or $\mathbb{P}^k$.  To reflect this generality, we use $\mathbb{P}^{k^ * }\in\\{\mathbb{P}^{\theta _ k^ * },\mathbb{P}^k\\}$ to denote the general learning target in the pseudo-code (Alg. 1).
>
> We have included a more detailed explanation in the revised manuscript to clarify this notation, and hope these additions resolve any ambiguity and make the framework and notation clearer.
>
> ---
>
> > **Weakness 1.3**: (16) not appear in the discrete objective
>
> In the continuous setting, we derive the weight $\frac{\mathrm{d}\mathbb{P}^{k^ * }}{\mathrm{d}\mathbb{P}^{\theta _ {k-1}}}$ through (14)–(15), where (15) shows that computing this weight reduces to evaluating the RN derivative $\frac{\mathrm{d}\mathbb{P}^\text{ref}}{\mathrm{d}\mathbb{P}^{\theta _ {k-1}}}$. Following the same logic, for the discrete case we explicitly provide the corresponding RN derivative $\frac{\mathrm{d}\mathbb{P}^\text{ref}}{\mathrm{d}\mathbb{P}^{\theta _ {k-1}}}$ in (16). Now, inserting RN derivative in (16) into the weight formula in (15) directly yields the desired importance weights for the discrete objective. To avoid confusion, we have revised the manuscript to clarify this in line 345-348.
>
> ---
> > **Weakness 2 & Question 4**: Comparisons regarding the cost of training. Is it $K$-times as expensive?
>
> Thanks for raising this important point. Although our method requires to solve $K$ proximal steps, as we have a good initialization $\mathbb{P}^{k-1}$ from the previous proximal step, we do not need to train each proximal step for as many iterations as training from scratch. Therefore, the overall training cost is comparable to training for the target distribution from scratch. For instance, LJ-55, Ising and Potts require 80, 31, and 20 hours on a single H100 GPU, respectively, which is comparable to the runtimes of baselines (e.g., ASBS [1]). Plus, due to the mode collapse issue for training for the untempered objective from scratch, directly training a single model would usually fail no matter how many steps we train. See, e.g., Fig. 1.

---

> > ### Author Response · Authors · 2025-11-21
> >
> > > **Weakness 3.1**: Hyperparameter sensitivity
> >
> > We appreciate the reviewer’s question regarding hyperparameter sensitivity. Because our method performs updates through a proximal step, the most influential hyperparameter is the proximal step size $\eta_k$, or equivalently the trust-region bound $\epsilon$. We provide a detailed analysis of this hyperparameter in Apps. D.3 (continuous target) and E.3 (discrete target).
> >
> > As shown in the App. D.3, for continuous targets, there is a clear sweet spot for $\epsilon$ in the range of $0.1 \leq \epsilon \leq 1$. A smaller $\epsilon$ enforces conservative updates, resulting in robust but slow convergence. Conversely, a larger $\epsilon$ accelerates convergence but can lead to instability or mode collapse. Our experiments demonstrate that the intermediate range achieves a favorable balance between stability and speed. For discrete targets, we found that the method is quite robust to the choice of $\epsilon$ ranging from $0.05$ to $5$.
> >
> > Aside from the proximal step size, we have not encountered any instabilities. Our method is also considerably less sensitive to hyperparameters than existing approaches (e.g. WDCE, MDNS), probably due to the stabilizing effect of the proximal update.
> >
> > ---
> >
> > > **Weakness 3.2 & 3.3**: Memoryless reference dynamics
> >
> > The reviewer is correct that our framework relies on memoryless reference dynamics. This assumption ensures that, when solving the optimization problem (3) over the space of path measures, the resulting optimal solution $\mathbb{P}^*$ attains the desired terminal distribution $\pi$. We have modified our proof of (3) in App. C.1 as well as added a remark after that to highlight the importance of this assumption, and refer readers to the adjoint matching paper [1] for further discussion.
> >
> > **Reference**: [1] Domingo-Enrich et al. Adjoint Matching: Fine-tuning Flow and Diffusion Generative Models with Memoryless Stochastic Optimal Control. ICLR 2025.
> >
> > ---
> > > **Question 1**: Energy histogram for the particle problems
> >
> > Thanks for the suggestion. For continuous target distributions, we have already included in Fig. 9. We also added the energy histogram for discrete target distributions in Figs. 19 and 20.
> >
> > ---
> > > **Question 3**: Extension to Molecular task
> >
> >
> > We thank the reviewer for this helpful suggestion. Following the reviewer’s advice, we have added experiments on the Alanine Dipeptide (AD) benchmark, following prior work such as ASBS [2]. Please see Sec. 5 for the detailed results. In this revision, we include quantitative comparisons with recent benchmarks, as well as torsion and Ramachandran plots for both ground-truth and generated samples.
> >
> > As demonstrated by both the quantitative and qualitative results, PDNS attains strong performance that is comparable to the current state-of-the-art method, ASBS [2]. However, the Ramachandran plot in fig. 10 also reveals that our model misses a few modes with extremely low probability mass. Further analysis shows that this issue originates from the initial reference samples generated by the initial path measure $\mathbb{P}^0$, which already exhibits mode collapse (see fig. 11). Because the proximal updates refine $\mathbb{P}^0$ in a progressive manner, modes absent in $\mathbb{P}^0$ may not be recovered during training. Detailed discussion is in Sec. 5 and App. D.4.
> >
> >
> > **Reference:** [2] Liu et al. Adjoint Schrödinger Bridge Sampler. NeurIPS 2025.
> >
> > ---
> >
> > We appreciate your positive evaluation of our work again, and we hope that our responses and the revised manuscript address your concerns.

---

> ### Comment · Reviewer_Kemc · 2025-11-25
> **improved notation & new application**
>
> I am very grateful for the clarifications and the addition of alanine dipeptide. While I am a bit disappointed that the method was unable to find the missing mode (very important in AD!) I still think the paper is strong. I will raise my confidence but keep my score.

---

> > ### Author Response · Authors · 2025-11-28
> >
> > We thank the reviewer for the comments on our newly included experiments. We will continue striving for developing more scalable and reliable neural sampling methods in the future.

---

### Official Review · Reviewer_ptdZ · 2025-11-01

**Soundness:** 3
**Presentation:** 3
**Contribution:** 2
**Rating:** 4
**Confidence:** 3

**Summary:**

This paper addresses general-purpose sampling by simulating diffusion inference trajectories in both continuous and discrete settings. A central challenge is that fitting the noisy target distribution with a neural network often leads to mode collapse and training instability. To mitigate these issues, the authors augment standard SOC objectives (e.g., CE, WDCE) with a KL-regularized local objective, yielding a proximal point method in path space. They instantiate this “path-measure proximal point” framework (PDNS) for both continuous SDEs and discrete CTMC diffusion samplers, demonstrating its generality.

**Strengths:**

1. The proximal-in-path-measure perspective is clean and broadly applicable across continuous SDE and discrete CTMC settings. In particular, the proximal WDCE naturally tempers importance weights, directly addressing mode collapse and high-variance gradients.
2. The paper presents the SOC, CE, WDCE, proximal formulations, and Girsanov/CTMC Radon–Nikodym weights with clarity. Practical guidance (e.g., OU reference bridges, conditional score formulas, and step-size schedulers) facilitates reproducibility.
3. Experiments span synthetic multimodal and high-dimensional mixtures, challenging particle systems (LJ-13/55), discrete Ising/Potts models near criticality, and combinatorial max-cut. PDNS achieves state-of-the-art or near-state-of-the-art results on multiple tasks, measured by domain-relevant metrics (Sinkhorn/MMD, $W_2$, energy $W_2$, magnetization, two-point correlations).

**Weaknesses:**

1. Proximal WDCE introduces overhead from replay buffers, weight computation (Girsanov/CTMC), and resampling. While results are strong, the paper does not quantify compute and memory costs relative to baselines (e.g., wall-clock, NFE, memory footprint), especially on the largest tasks (LJ-55, Potts). Resource-normalized comparisons would clarify efficiency.
2. Many derivations rely on independence between initial and terminal distributions (Eq.~(2)). The framework assumes (nearly) memoryless references to obtain closed forms and reciprocal sampling, which may be unavailable for some targets. Although non-memoryless extensions are mentioned in related work, PDNS is not analyzed under such references.
3. The proximal step size $\eta_k$ (or $\lambda_k$) critically trades off stability and speed. Despite an adaptive KL-based scheduler, the paper lacks systematic sensitivity studies and practical heuristics for choosing $\epsilon$ across tasks and scales. Additional ablations on $\eta_k$, buffer size, and resampling vs.\ weighting would aid practitioners.

If the authors resolve my concerns, I am willing to raise my rating.

**Questions:**

1. Did authors consider using the Wasserstein two distance as a regularizer to replace the KL divergence between $p^{\theta}$ and $p^{\theta_{k-1}}$, by coupling the JKO scheme, the proximal problem formulation may be more intuitive.

---

> ### Author Response · Authors · 2025-11-21
>
> We sincerely thank the reviewer for carefully reading our paper and providing valuable feedback. Please see our response below.
>
> > **Weakness 1**: Quantifying compute and memory/runtime costs relative to baselines
>
> We thank the reviewer for this important suggestion. We summarize our computational costs below.
>
> All training hyperparameters for continuous target distributions, including the number of function evaluations (NFEs) for simulating the SDE, are reported in Tab. 8. For discrete settings, we have also described the training hyperparameters in App. E.2.2. For MDM, as we always use random-order autoregressive sampling, the NFE for sampling is equal to the length of the sequence. In terms of wall-clock time, PDNS requires approximately 80 hours on the LJ-55 potential when trained on a single NVIDIA H100 GPU, which is comparable to the 80 hours for ASBS [1] under same conditions. The training for $24\times24$ Ising model and $16\times16$ Potts model complete in around 31 and 20 hours, respectively, on a single NVIDIA H100 GPU. Our target distributions have larger dimension than MDNS [2] and thus requires longer running time.
>
> Because our approach uses identical neural architectures to Adjoint Sampling [3] (for continuous diffusion) and MDNS [2] (for discrete diffusion, except with a larger embedding dimension due to larger size of target distribution), the GPU memory footprint is similar across methods. Specifically, training on LJ-55, $24\times24$ Ising and $16\times16$ Potts consume approximately 14, 70, and 37 GB GPU memory, respectively.
>
> Furthermore, our replay buffer stores final samples (without full generation trajectories) from the control policy obtained from previous proximal step, and these samples are remained the same during each proximal step. So, buffer serves solely as a simple sample cache. The maximum buffer size is approximately 50K samples, corresponding to under 400MB of memory usage, a minor portion of the overall training memory footprint. Additionally, the resampling step introduces negligible computational overhead, typically requiring less than 10 seconds per proximal iteration.
>
> > **Weakness 2**: Reliance on memoryless reference dynamics
>
>
> We appreciate the reviewer's insightful comment. We agree that extending our proximal method to non-memoryless reference dynamics would be a compelling future direction. However, such an extension is non-trivial, as it would require adapting the control formulation introduced in ASBS [1], which learn both a control and a corrector term to address non-memoryless reference. Moreover, for discrete case, as we employ the *masked* diffusion, the process is automatically memoryless since the initial distribution is always a delta distribution. While this direction of research is interesting, an extension to non-memoryless references lies beyond the scope of this paper.
>
> We would also like to emphasize that the memoryless reference assumption does not restrict the range of target distributions to which PDNS can be applied. Since the reference dynamics is entirely decoupled from the target distribution, we are free to construct a memoryless reference process in any sampling problem. Given any memoryless reference dynamics, PDNS learns the control to sample the target distribution.

---

> > ### Author Response · Authors · 2025-11-21
> >
> > > **Weakness 3**: Lack of systematic sensitivity studies, practical heuristics, and ablation studies
> >
> > As the reviewer correctly pointed out, the proximal step size plays a critical role in determining the performance, stability, and convergence speed of our algorithm. In App. D.3 (continuous target) and E.3 (discrete target), we conduct extensive studies on the influence of various hyperparameters involved in the proximal updates, including:
> >
> > - By comparing the non-proximal and proximal methods, we demonstrate that the proximal approach shows improved stability and better performance.
> > - We ablate on across various schedules for $\epsilon$, highlighting the trade-off between stability and convergence speed.
> > - We compare an adaptive scheduler and fixed scheduler, showing the adaptive KL control provides a more robust balance across tasks compared to fixed scheduling (in continuous target).
> >
> > As the reviewer mentioned, too tight bound (small $\epsilon$) would stabilize the training, however, reduces the convergence speed, and looser bound may result in mode collapse.
> >
> > Furthermore, as the reviewer suggested, we add the resampling vs. weighting result as Tab. 7 in App. B. As shown in the table, reweighting and resampling achieve comparable performance in terms of the 2-Wasserstein distance $\mathcal{W} _ 2$. However, resampling consistently yields better performance on the energy-weighted 2-Wasserstein metric $E(\cdot) ~ \mathcal{W} _ 2$ in all tasks, albeit at the cost of slower convergence when using the adaptive scheduler.
> >
> > Additionally, in all of our experiments, we use a buffer size of approximately 50k, which is sufficiently large to ensure stable performance. Since our method consistently performs well under this setting, we observe that as long as the buffer size is reasonably large, the results remain stable. Therefore, we do not regard the buffer size as a critical hyperparameter in our method.
> >
> > ----
> >
> > > **Question 1**: Replacing the KL divergence proximal term to the Wasserstein-2 distance
> >
> > We appreciate the reviewer’s thoughtful suggestion regarding the possibility of replacing the KL-based proximal term with a Wasserstein-2 (W2) regularizer. This is indeed an interesting direction. Given the current iterate $\mathbb{P}^{\theta _ {k-1}}$, the Wasserstein-based update takes the form
> > $$\mathbb{P}^{\theta _ k^ * }=\operatorname{argmin} _ {\mathbb{P}^\theta}\left(-\mathbb{E} _ {\mathbb{P}^\theta(X)}r(X _ T)+\operatorname{KL}(\mathbb{P}^\theta\|\mathbb{P}^\text{ref})+\frac{1}{\eta _ k}W _ 2^2(\mathbb{P}^\theta,\mathbb{P}^{\theta _ {k-1}})\right)$$
> > While conceptually appealing, we believe that there are two main challenges in implementing this idea:
> >
> > - First, unlike the KL divergence where we could obtain a closed-form expression for the RN derivative $\frac{\mathrm{d}\mathbb{P}^{\theta _ k^ * }}{\mathrm{d}\mathbb{P}^{\theta_{k-1}}}$ (eq. (14-16)), in the case of W2 distance, we do not have such a closed-form expression. This makes it difficult to use the WCE or WDCE losses, as they require this RN derivative to do resampling or reweighting (see Sec. 2).
> >
> > - Second, if we directly treat the right-hand side of the above equation as the training loss, then we need to estimate the W2 distance between two path measures $\mathbb{P}^\theta$ and $\mathbb{P}^{\theta _ {k-1}}$ at each iteration, which is computationally expensive and non-trivial.
> >
> > Therefore, while using the W2 distance as a proximal term is an interesting idea, it may require significant modifications to our current framework to address these challenges effectively.
> >
> > ---
> >
> > Thank you again for your valuable feedback, and we earnestly hope that our responses and the revised manuscript address your concerns. Please feel free to reach out with any further questions or comments.
> >
> >
> > **References**
> >
> > [1] Liu et al. Adjoint Schrödinger Bridge Sampler. NeurIPS 2025.
> >
> > [2] Zhu et al. MDNS: Masked Diffusion Neural Sampler via Stochastic Optimal Control. NeurIPS 2025.
> >
> > [3] Havens et al. Adjoint Sampling: Highly Scalable Diffusion Samplers via Adjoint Matching. ICML 2025.

---

### Author Response · Authors · 2025-11-21
**General Response**

We would like to convey our sincere gratitude to all reviewers for carefully reading our paper and providing thoughtful comments and constructive feedback. We are glad to hear that the reviewers acknowledged the novelty (h5wm) and contribution (Kemc) of this work, *unanimously* agreed on the general applicability of our proposed methodology, commented that the paper is clean (ptdZ, prWn), easy to follow (Kemc, prWn), and well-motivated (h5wm) for solving a common practical issue (Kemc). Finally, the reviewers found most of the experimental results convincing (ptdZ, prWn, h5wm). Our paper proposed a unified framework for stable and efficient training of diffusion-based neural samplers through proximal gradient descent in the space of path measures, which we believe would be of significant interest to the community, and hopefully, could open up new avenues for future research.

The main concerns raised by the reviewers are summarized as follows:

1. Insufficient quantification of the computational complexity of the proposed method, due to the outer loops for solving each subproblem. (ptdZ, Kemc)

2. Complexity of notations regarding the path measures and insufficiency of explanations that may pose challenges for the general audience. (Kemc, h5wm)

Moreover, the reviewers ptdZ and Kemc also raised questions regarding the memoryless assumption on the reference path measure, which is an interesting point for further discussion and future work.

We will address specific concerns raised by the reviewers point-by-point in the individual response. Please also see the updated manuscript for changes made in response to the reviews, which are highlighted in $\color{orange}{{\rm orange}}$ color. Notable changes include:

1. We included further experimental results and ablation studies in Apps. B (weight-based v.s. resampling-based realization of WDCE), Sec. 5 (alanine dipeptide), and E.3 (ablation studies on discrete target distributions).

2. We added a table after Prop. 3.1 to summarize the meaning of different path measures used in the paper, and polished the related discussions in Secs. 3 and 4 for better readability.

3. We modified the proof of (3), the construction of optimal path measure $\mathbb{P}^*$, and included a discussion to highlight the purpose of the memoryless assumption of the reference path measure $\mathbb{P}^\text{ref}$.

We sincerely hope that the revisions and responses adequately address the reviewers' concerns and show our commitment to improving the quality of our work. We also welcome any further questions or comments from the reviewers.

---

### Author Response · Authors · 2025-12-02
**Rebuttal Overview -- Key Contributions, Concerns, Responses, and Updates**

We thank all reviewers again for their valuable feedback and constructive comments, and we also thank PCs, SACs and ACs for their time and effort in managing the review process in face of the OpenReview system issues during the rebuttal period. As the reviewers cannot engage in further discussions anymore at the current stage, we summarize the main points of our rebuttal here for clarity for the ACs in hope that it can help with the final decision.

# Contribution and Novelty of our Work

Our paper studies the task of learning a diffusion-based neural sampler for complex target distributions, a fundamental problem with broad applications in machine learning and scientific computing. We mainly focus on addressing the mode collapse issue of the stochastic optimal control (SOC)-based neural samplers, which arises when directly training from scratch to fit the untempered target distribution. Our main contributions and novelties are as follows:

- **Unifying framework for SOC-based neural samplers**: We integrate both continuous and discrete SOC-based samplers into a unified framework through path measures, and introduce Proximal Diffusion Neural Sampler (PDNS) that performs proximal update in path space to stabilize learning and mitigate mode collapse.
- **Novel training objectives**: We develop **proximal weighted denoising cross-entropy** (proximal WDCE) for both continuous and discrete settings as a concrete example of PDNS, which is scalable and memory-efficient.
- **Superior empirical performance**: We provide extensive experiments across continuous and discrete sampling tasks, demonstrating that PDNS improves stability and sampling quality compared with strong baselines.

We believe that our work makes significant progress to the field of neural sampling, and would inspire future research in this direction.

# Reviewers' Main Concerns and our Responses

We are grateful to the reviewers for their careful reading and constructive comments. Below, we summarize the representative concerns raised by the reviewers, and briefly outline our responses. Detailed point-to-point responses have been provided in individual responses to each reviewer.

## (I) Methodological questions

The reviewers raised questions regarding

1. Whether the framework can be applied to **non-memoryless** reference dynamics (ptdZ, Kemc);
2. Explanation, clarification, and simplification of the **path measure notations** and related derivations (Kemc, h5wm);
3. The connection with the related work **Blessing et al., NeurIPS 2025** (arxiv:2508.12511) (prWn).

In our rebuttal, we have incorporated the following explanations and updates into the paper to address these concerns:

1. We have added further discussion on the memoryless assumption of the reference dynamics in App. C.1 to clarify its role in ensuring the optimal path measure **attains the desired terminal distribution**, while we also acknowledge the non-trivial challenges in extending to non-memoryless references as future work.
2. We have added Tab. 1 under Prop. 3.1 to **summarize the meaning of different path measures** used in the paper, and polished the related discussions in Secs. 3 and 4 for better readability.
3. We have added a detailed comparison with Blessing et al. in the response to Reviewer prWn, highlighting the key differences in terms of the **objective functions**, **algorithms**, **memory requirements**, and **applicability to discrete settings**.

## (II) Experiment-related questions

The reviewers asked us to

1. Provide quantification of the **computational complexity** of PDNS as it involves solving multiple proximal subproblems (ptdZ, Kemc);
2. Include **ablation studies** on hyperparameters and design choices (ptdZ);
3. Try the method on **more complex target distributions** such as alanine dipeptide (Kemc).

In response,

1. We summarize the **training hyperparameters and computational cost** in Apps. D.2.3 and E.2.2 for continuous and discrete target distributions, respectively, showing the overall training cost is comparable.
2. We include more extensive **ablation studies**, including: weight-based v.s. resampling-based realization of WDCE in App. B and ablation studies on discrete targets (prefixed and adaptive schedulers) in App. E.3. Ablation studies on continuous targets (with or without proximal step, effect of trust region bound $\epsilon$, choice of the scheduler) are already included in App. D.3 during the initial submission.
3. We have added experiments on **alanine dipeptide** in Sec. 5.1, demonstrating it achieves comparable performance to the state-of-the-art method ASBS.

In addition to our response to the reviewers' comments, we have also made significant improvements to the paper during the rebuttal period. Please refer to the general response part below for detailed explanations, and check our updated manuscript where notable changes are highlighted in $\color{orange}{\rm orange}$.

---

> ### Author Response · Authors · 2025-12-02
>
> # Concluding Remarks
>
> We thank the reviewers and ACs again for their time and effort in reviewing our paper and managing the review process. We hope that our responses and improvements during the rebuttal period have adequately addressed the concerns raised by the reviewers. We will continue to improve the paper based on the valuable feedback received.

---

### Meta-Review · Area_Chair_XToB · 2026-01-06

**Summary:**

Reviewers were generally positive about the paper’s core contribution: a proximal point framework in path-measure space (PDNS) that unifies SOC-based neural samplers for both continuous and discrete settings and improves training stability. The main concerns focused on clarity and notation, computational and memory overhead, the reliance on a memoryless reference dynamics assumption, and the breadth of experimental validation, including requests for ablations and more complex targets. The rebuttal and revised manuscript addressed these points by clarifying notation and objectives, adding explicit compute/memory reporting, expanding ablation studies, and including additional diagnostics and experiments. Several reviewers explicitly noted that their concerns were resolved, and at least one reviewer increased their score, leaving the remaining issues as natural directions for future work rather than blockers, which supports an Accept recommendation.

**Reviewer Concerns:**

- Reviewer ptdZ: Main concerns were (i) quantifying compute/memory/runtime overhead of solving multiple proximal subproblems, (ii) sensitivity/ablations for the proximal step size / scheduler and buffer/resampling choices, and (iii) limitations from the (nearly) memoryless reference assumption.

- Reviewer Kemc: Main concerns were clarity of the exact training objective and path-measure notation (including RN-derivative terms), plus practical cost versus baselines, hyperparameter sensitivity, and a request to validate on a more realistic molecular benchmark (e.g., alanine dipeptide).

- Reviewer prWn: Main concerns were whether the experiments convincingly demonstrate mitigation of mode collapse and how PDNS differs from closely related concurrent work (Blessing et al., 2025), including applicability to discrete settings.

- Reviewer h5wm: Main concerns were readability for less-experienced readers and stronger empirical evidence on more complex/high-dimensional targets, along with minor technical clarifications (path space definition) and fairness/details of LEAPS comparisons.

**Reviewer Scores:**

- Reviewer ptdZ: Likely increased or at least became more positive, since they explicitly indicated willingness to raise the rating if compute/memory and ablation concerns were addressed, which the rebuttal/revision directly did.

- Reviewer Kemc: Unchanged, as the reviewer explicitly said they would raise confidence but keep the score despite appreciating the added alanine dipeptide experiment and clarification.

- Reviewer prWn: Unchanged, as the reviewer explicitly stated their questions were addressed and that they would keep the score at 8.

- Reviewer h5wm: Increased, as the reviewer explicitly said the revised manuscript addressed concerns, improved readability, and added experiments, and they raised their score.

---

### Decision · Program_Chairs · 2026-01-26

Accept (Poster)